# Full-color, time-valve controllable and Janus-type long-persistent luminescence from all-inorganic halide perovskites

Tianhong Chen[1] & Dongpeng Yan [1]✉

Long persistent luminescence (LPL) has gained considerable attention for the applications in decoration, emergency signage, information encryption and biomedicine. However, recently developed LPL materials – encompassing inorganics, organics and inorganic-organic hybrids – often display monochromatic afterglow with limited functionality. Furthermore, triplet exciton-based phosphors are prone to thermal quenching, significantly restricting their high emission efficiency. Here, we show a straightforward wet-chemistry approach for fabricating multimode LPL materials by introducing both anion (Br⁻) and cation (Sn²⁺) doping into hexagonal $CsCdCl_3$ all-inorganic perovskites. This process involves establishing new trapping centers from $[CdCl_{6-n}Br_n]^{4-}$ and/or $[Sn_{2-n}Cd_nCl_9]^{5-}$ linker units, disrupting the local symmetry in the host framework. These halide perovskites demonstrate afterglow duration time ( > 2,000 s), nearly full-color coverage, high photoluminescence quantum yield ( ~ 84.47%), and the anti-thermal quenching temperature up to 377 K. Particularly, $CsCdCl_3$:$x$%Br display temperature-dependent LPL and time-valve controllable time-dependent luminescence, while $CsCdCl_3$:$x$%Sn exhibit forward and reverse excitation-dependent Janus-type luminescence. Combining both experimental and computational studies, this finding not only introduces a local-symmetry breaking strategy for simultaneously enhancing afterglow lifetime and efficiency, but also provides new insights into the multimode LPL materials with dynamic tunability for applications in luminescence, photonics, high-security anti-counterfeiting and information storage.

Long-persistent luminescence (LPL) is an intriguing optical phenomenon characterized by sustained luminescence for durations ranging from seconds to several days after the cessation of excitation. Its earliest documented observation date back to the 17th century[1]. However, significant advancements in this research field only materialized in the 20th century, notably with the discovery of LPL in copper-doped zinc sulfide, leading to its application in glow-in-the-dark materials[2]. A groundbreaking moment occurred in the mid-1990s

when Matsuzawa et al. introduced the green inorganic LPL material $SrAl_2O_4$:$Eu^{2+}$, $Dy^{3+}$, utilizing oxygen-vacancy traps[3]. Since then, a diverse array of afterglow phosphors, involving oxides, sulfides, and nitrides doped with various lanthanides or transition metals has been developed[4]. These materials have widespread applications in lighting[5], displays[6], bioimaging[7,8], photocatalysis[9], information storage and security encryption[10–12]. However, during this rapid expansion[13], scientists recognized that the high-temperature dry synthesis process

[1]Beijing Key Laboratory of Energy Conversion and Storage Materials, and Key Laboratory of Radiopharmaceuticals, Ministry of Education, College of Chemistry, Beijing Normal University, Beijing 100875, P. R. China. ✉e-mail: yandp@bnu.edu.cn

(1000 ~ 1500 °C) not only fails to meet the requirements of energy conservation and environmental protection requirements, but also poses a significant safety risk for manufacturers.

In response to these challenges, new distinctive design concepts have recently emerged, involving molecule-based LPL through chemical synthesis and/or molecular self-assembly. In 2017, Adachi et al. utilized two simple organic molecules to achieve LPL by recombining long-lived charge-separated states, marking the advent of organic LPL (OLPL)[14]. In 2022, Tang et al. disrupted the pattern of multi-component synergies by employing a single-component molecular LPL system capable of detectable afterglow for more than 12 min under ambient conditions[15]. Our group has contributed to this field by developing LPL systems based on organic-inorganic halides[16,17], which have emerged as promising and cost-effective semiconductor materials for sensor, optical waveguide, and information storage[18]. However, these hybrid perovskites have only exhibited short afterglow times, attributed to mechanisms such as room temperature phosphorescence (RTP) and thermally activated delayed fluorescence (TADF)[19]. In 2021, Zhang et al. reported a double halide perovskite system, $Cs_2Na_xAg_{1-x}InCl_6:y\%Mn$[20], incorporating energy transfer (ET) processes and self-trapped excitons (STEs) mechanisms to obtain LPL. Despite significant efforts in the aforementioned progress, achieving high luminescent efficiency in halide perovskite engineering or OLPL systems remains a formidable task[21]. Notably, the majorities of LPL materials tend to exhibit monochromatic (solitary color) afterglow, lacking proficiency in multifarious stimulated-response skills.

The recent extensive exploration of stimuli-responsive luminescent materials[22,23] underscores their excitation wavelength-dependence (Ex-De)[24], as well as their intelligent response to mechanical force[25], pH[26], electric field[27], and temperature[28] in various application scenarios. In addition to advancing the development of diverse molecules and manipulating their lifetimes and emission efficiency, it is crucial to establish a versatile platform for LPL materials to ensure their practical utility. Simultaneously, the burgeoning field of materials exhibiting time-dependent, color-varying afterglow holds promising prospects in optoelectronic devices and high-end anti-counterfeiting products[29]. In this context, two primary strategies exist for creating such exquisite materials. One involves incorporating fluorescent dyes as acceptors (guests) into a rigid polymer matrix donor (host), facilitating phosphorescence resonance energy transfer (PRET) in the host-guest system[30]. Commonly, their color-varying afterglow is shifted from long to short wavelengths, while the occurrence of afterglow changing towards longer wavelengths is a rare and huge task. The other entails constructing multiple luminescence centers through the regulation of the triplet and singlet energy levels[31]. Nevertheless, these advanced schemes have several drawbacks, including potential cross-chromaticity with multiple similar fluorescent dyes, short lifetimes at the millisecond to second level, challenges in tailoring a single component, and difficulty in controlling the discoloration time point during the afterglow process. Notably, the controllable time valve at the color change point is a significant gap in this field. To overcome these challenges through halide perovskite engineering, several requirements must be met: (a) achieving ultralong persistent luminescence, (b) demonstrating multimode luminescence, (c) exhibiting a wide range of afterglow color variability, (d) allowing for easy determination by the naked eye, and (e) enabling an adjustable time valve for afterglow discoloration based on specific variables. In the pursuit of performance breakthroughs for typical $ABX_3$ all-inorganic perovskites, the focus primarily revolves around the regulation of B or X sites[32].

In this work, we propose that introducing doped ions into the all-inorganic skeleton can disrupt the original symmetry, forming new trap states and luminescence centers. Here, we present a dually positive design strategy to achieve color-tunable LPL by introducing $Br^-$ or $Sn^{2+}$ ions into the hexagonal phase $CsCdCl_3$ through a modified wet-chemistry method. This involves (a) ensuring the $Br^-$ or $Sn^{2+}$ ion radius is comparable to that of $Cl^-$ and $Cd^{2+}$ ions, (b) leveraging the $4p$ orbital effects of $Br^-$ ions on the bandgap and the $5s^2$ electronic configuration of $Sn^{2+}$ ions to distort the lattice, and (c) using doping to break the local symmetry in the main framework, thereby establishing different trapping centers to compensate for forbidden energy transitions. Our findings indicate that disrupting geometric symmetry may generate multimode luminescence in $Br^-$ or $Sn^{2+}$-doped perovskites at both face-shared ($C_{3v}$ symmetry) and corner-shared ($D_{3d}$ symmetry) $[CdCl_6]^{4-}$ octahedrons. Thermoluminescence (TL) curves demonstrate the coexistence of shallow and deep trapping centers in both $Br^-$ or $Sn^{2+}$-doped perovskites, contributing to their anti-thermal quenching ability up to 377 K. Ultimately, $CsCdCl_3:x\%Br$ and $CsCdCl_3:x\%Sn$ exhibit long afterglow durations (2000 s), with optimized photoluminescence quantum yields (PLQY) of 84.47% and 65.71%, respectively, representing cutting-edge levels among current LPL perovskites and inorganic-organic hybrids. Significantly, $CsCdCl_3:x\%Br$ demonstrates remarkable color-varying long-afterglow properties, with color alteration at different time points precisely regulated by varying concentrations of $Br^-$ ions. Moreover, $CsCdCl_3:x\%Br$ displays wide-range (97–377 K) temperature-dependent PL properties, enabling full-color adjustability. Specifically, $CsCdCl_3:x\%Sn$ exhibits a unique optical behavior analogous to Janus-type emission[33], including forward and reverse excitation-dependent LPL at low or room temperature, respectively. These multifunctional LPL perovskites hold substantial potential for high-level anti-counterfeiting and information security in extreme scenarios.

## Results

### Synthesis and structure

A straightforward synthesis method is essential for achieving LPL. In the elevated temperature dry synthesis method (1000 ~ 1500 °C) for afterglow phosphors[34], the high-temperature melting process towards OLPL materials heavily depends on the melting/boiling point similarity of each component to prevent bond breakage and reorientation[35]. Conventional solution chemistry methods have been employed for the short-lived afterglow of organic-inorganic halides and crystalline/polymeric organic materials[19,20,36]. In this work, the crystals of $CsCdCl_3$ can be grown using a modified hydrothermal reaction[37,38] (details in "Methods", Fig. 1f), and its crystal lattice adapts a space group P6_3/mmc (CCDC No. 2313854, Supplementary Table 1). The 3D asymmetric unit, as shown in Fig. 1a and Supplementary Fig. 1, is constructed with $[CdCl_6]^{4-}$ octahedrons. Two of these share a triangular face to form $[Cd_2Cl_9]^{5-}$ in $C_{3v}$ symmetry, which then connected with six additional $[CdCl_6]^{4-}$ octahedra to achieve corner-shared $D_{3d}$ symmetry. This unique packing arrangement offers numerous coordination sites for diverse halides and divalent metal cations, allowing for the arbitrarily anchoring of $Br^-$ or $Sn^{2+}$ ions at the $Cl^-$ or $Cd^{2+}$ ion sites, potentially leading to distinct optical properties. X-ray photoelectron spectroscopy (XPS) profiles describe $Br^-$ and $Sn^{2+}$-doped samples, with the characteristic peaks of $Br^-$ $3d$ and Sn ion $3d_{3/2}$ and $3d_{5/2}$ becoming more pronounced with increasing guest-doped concentration (Supplementary Figs. 2, 3). Particularly, the peaks centered at $3d_{3/2} = 496.02$ eV and $3d_{5/2} = 487.02$ eV correspond to $Sn^{2+}$[39,40], suggesting that a tiny amount of $Sn^{2+}$ doping can preserve its stability (Supplementary Fig. 3b, d).

Upon alternation by $Br^-$ or $Sn^{2+}$ ions, the powdered X-ray diffraction (PXRD) patterns of $CsCdCl_3:x\%Br$ and $CsCdCl_3:x\%Sn$ closely agree with the pattern in the PDF#18-0337, confirming the single-phase purity of the synthesized $Br^-$ or $Sn^{2+}$-doped $CsCdCl_3$ (Fig. 1b, c and Supplementary Figs. 4, 5). The ion radius values of $Br^-$ ($r = 1.96$ Å) and $Sn^{2+}$ ($r = 1.02$ Å, CN = 6) are larger than those of $Cl^-$ ($r = 1.81$ Å) and $Cd^{2+}$ ($r = 0.95$ Å, CN = 6), respectively, contributing to the expansion of the host lattice, manifested by the shifting of Bragg positions at [104] and [110] to lower angles (Supplementary Figs. 4, 5). From these images of

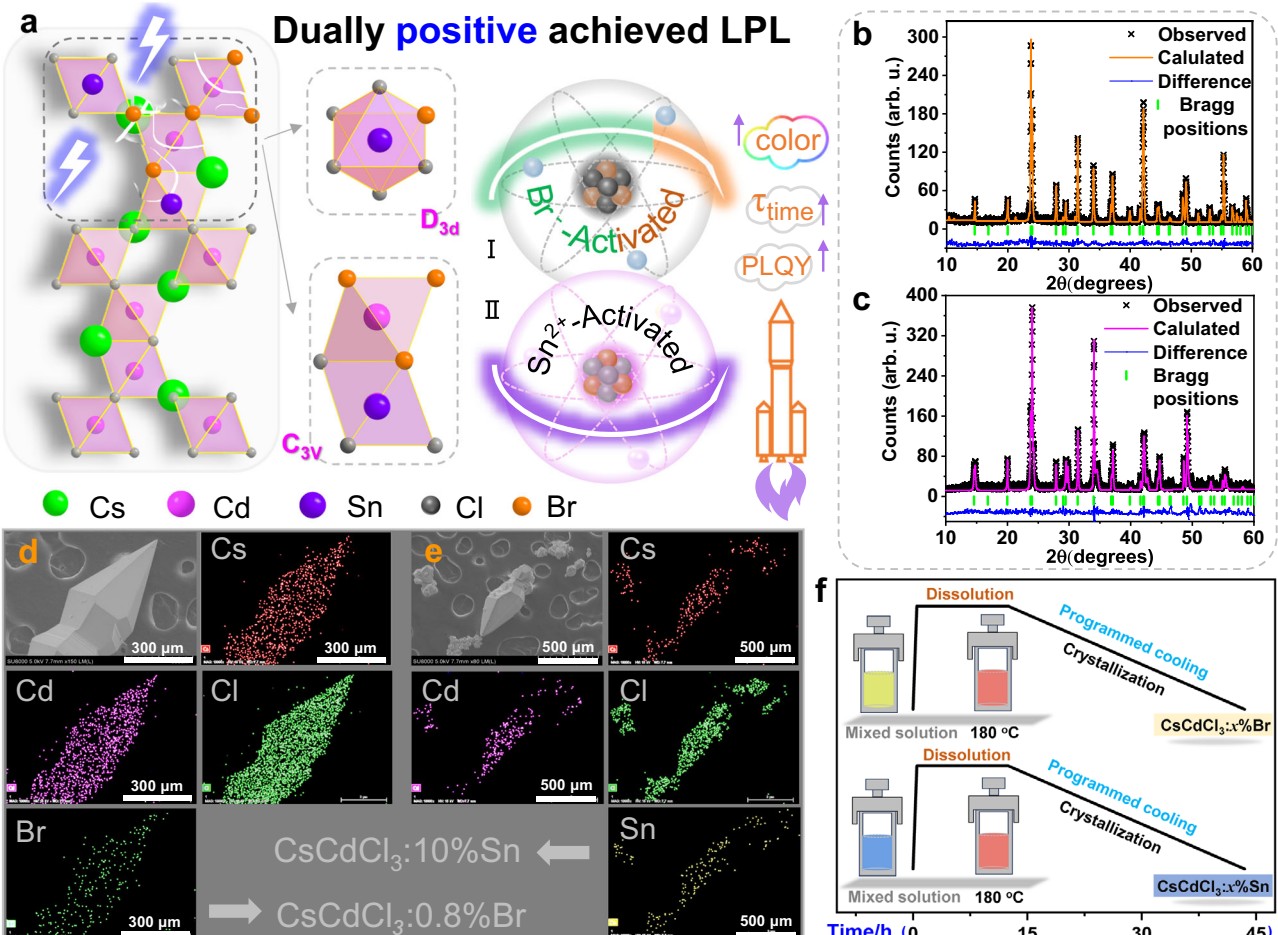

**Fig. 1 | Design concept and material characterization. a** Schematic representation of the rational design of Cl⁻ or Cd²⁺ sites in the CsCdCl₃ crystal structure, occupied by (I) Br⁻ or (II) Sn²⁺ ions to disrupt local symmetry in the host framework and promote LPL, color and PLQY properties. Details: The design employs shadows and lighting to highlight the inherent CsCdCl₃ crystal structure was disrupted by activated Cl⁻ or Sn²⁺ ions, bringing out a dual positive performance improvement strategy. Rietveld refinements of the typical XRD patterns of **b** CsCdCl₃:0.8%Br and **c** CsCdCl₃:10%Sn. SEM image of **d** CsCdCl₃:0.8%Br and **e** CsCdCl₃:10%Sn, along with their corresponding elemental mapping images of Cs, Cd, Cl, Br and Sn. **f** Wet-chemistry method for the synthesis of CsCdCl₃:x%Br and CsCdCl₃:x%Sn all-inorganic perovskites.

as-synthesized crystals (Supplementary Figs. 6, 7), all CsCdCl₃:x%Br and CsCdCl₃:x%Sn exhibit remarkably consistent and pure fluorescence colors, also demonstrating the uniformity and single-phase purity in this doping engineering. Scanning electron microscope (SEM) images show a typical spindle shape of Br⁻ or Sn²⁺-doped CsCdCl₃ crystals (Supplementary Fig. 8), with uniform distribution of elemental constituents (Cs, Cd, Cl, Br or Sn) in element mapping images, confirming the successful dopant engineering of Br⁻ or Sn²⁺ ions (Fig. 1d, e). The standard Rietveld refinement technique reveals a poor linear relationship between the nominal concentrations of Br or Sn and the distance of the (110) planes, somewhat deviating from the Vegard's law. To investigate this deviation, energy dispersive spectroscopy (EDS) was employed to determine the actual concentrations of Br⁻ or Sn²⁺ ions in crystals (Supplementary Figs. 4b, 5b and Supplementary Tables 2, 3). Interestingly, the actual concentrations exhibit perfect linearity with respect to the $d_{110}$ plane and strictly adhere to Vegard's law[41]. Furthermore, the ICP-OES value of the doping Sn²⁺content in CsCdCl₃:10%Sn is 7.01% (Supplementary Table 4), which is similarly to the corrected Vegard's law (XRD) and EDS values. These results suggest that the concentrations of Br-doping slightly exceed the nominal value, while the opposite holds true for Sn-doping, which can be attributed to differences in solubility and solvent boiling points.

## Photophysical Properties

The optical characteristics of CsCdCl₃ single crystals were initially investigated. As depicted in Supplementary Fig. 9a–c, the optimal excitation wavelength for the photoluminescence excitation (PLE) center of pure CsCdCl₃ is 254 nm, inducing a broad emission peak at 595 nm with a full width at half-maximum (FWHM) of 88 nm in both prompt and delayed spectra. CsCdCl₃ displays robust excitonic absorption (Supplementary Fig. 10a), aligning well with the above-mentioned PLE spectrum. Given the substantial Stokes shift (341 nm) and wide FWHM, the observed orange emission is attributed to the self-trapped excitons (STEs) emission, consistent with the prior report[42]. Nevertheless, the low emission intensity (PLQY~25.47%) significantly restricts its applicability (Fig. 2i and Supplementary Fig. 11a).

One inspiration for multi-color LPL is the promise of employing different halide cations with tunable bandgaps, as well as incorporating halogen substitution in lead-based halide perovskites, enabling the full-color photoluminescence within nanosecond lifetimes[43]. As shown in Fig. 2a and Supplementary Fig. 9a, under 254 nm excitation, CsCdCl₃:x%Br (x = 0.2–15) crystals exhibit a blue-shifted stronger and broader emission peak at 482 nm compared to the pristine CsCdCl₃ crystals. The delayed spectra show emission peaks at 482 and 595 nm are both enhanced with increasing Br⁻ doping concentration (Fig. 2a

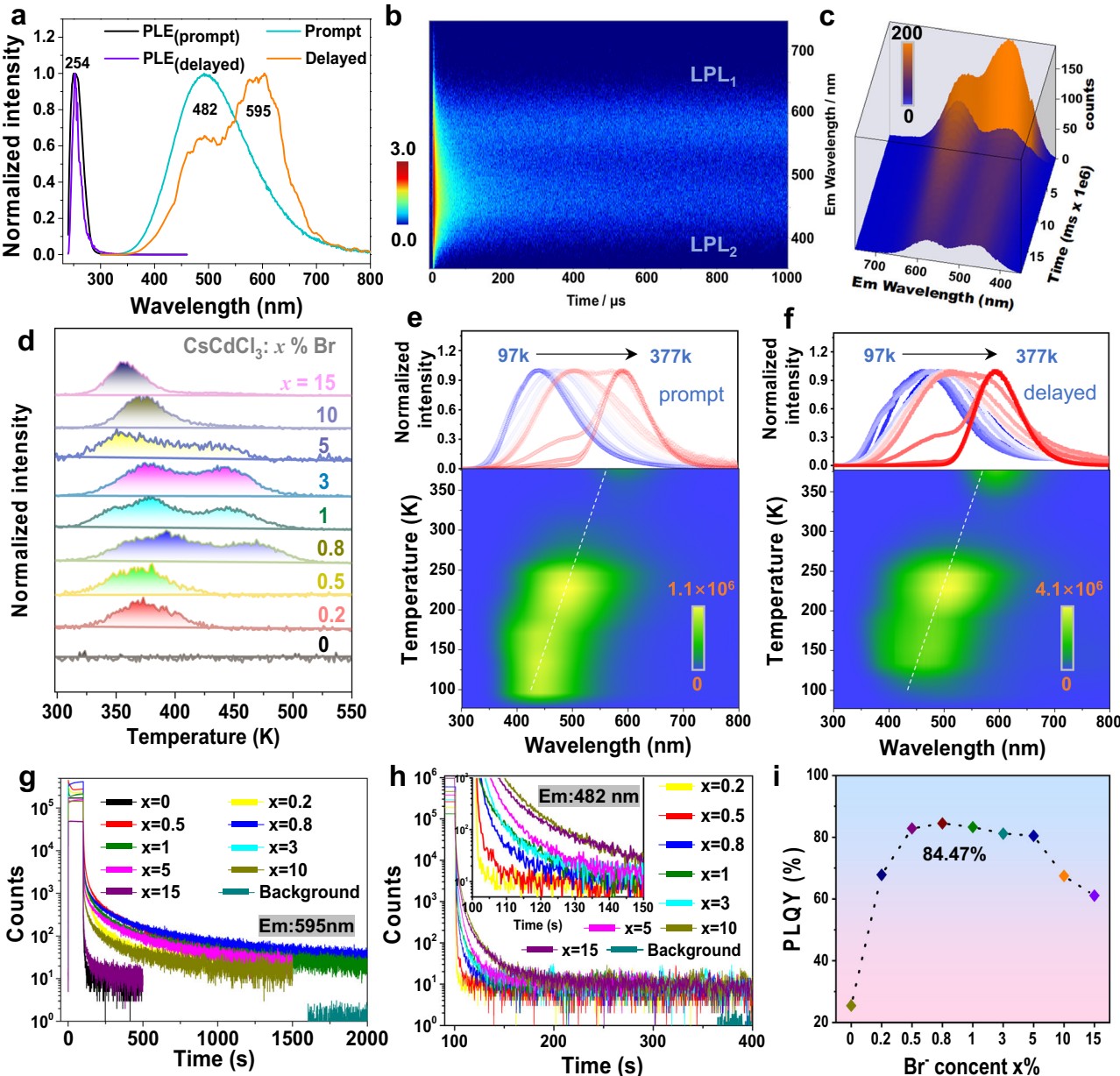

**Fig. 2 | Photophysical properties of CsCdCl₃:x%Br. a** Normalized PL spectra of CsCdCl₃:0.8%Br based on prompt and delayed (t_d = 1 ms) patterns under 254 nm excitation, along with the corresponding normalized PLE spectra. **b** Pseudo color map of time-resolved PL spectra of afterglow for CsCdCl₃:0.8%Br under 254 nm flash lamp. **c** Three-dimensional time-resolved PL spectra of CsCdCl₃:0.8%Br under 254 nm excitation. **d** TL spectra of CsCdCl₃:x%Br were obtained after pre-irradiation with a 254 nm UV lamp for 1 min. Temperature-dependent PL mapping of CsCdCl₃:0.8%Br based on **e** prompt and **f** delayed (t_d = 1 ms) patterns under 254 nm excitation. Afterglow decay curve of CsCdCl₃:x%Br with the detected emission wavelength at **g** 595 nm and **h** 482 nm. **i** The PLQY values of CsCdCl₃:x%Br. Note: Due to the limited space of the picture, the color gradient of curves in Fig. 2e, f only represents the change in emission peaks based on temperatures (the arrows from left to right: 97, 117, 137, 157, 177, 197, 217, 257, 277, 297, 317, 357, and 377 K, respectively), without any special meaning.

and Supplementary Fig. 9b), revealing that these two peaks merge to a broaden peak in the prompt spectra. CsCdCl₃ possesses both C₃ᵥ and D₃d symmetries[44], where the a₁ → e transition allowed in C₃ᵥ symmetry and the a₁g→eg transition in D₃d symmetry require undergoing an S-T-splitting route[45]. The radiative transition in both symmetries originates from the triplet exciton, with the energy gap of D₃d tending to be larger than that of C₃ᵥ[45]. Previous studies have indicated that D₃d symmetry's PL is in the UV region at low temperatures due to constrained molecular vibrations accelerating the S-T splitting process[45,46]. However, CsCdCl₃:x%Br (x = 0.2–15) crystals show robust emission centered at 482 nm without the need for low temperatures. To comprehend this behavior, we analyze the structure-luminescence relationship, the

newly formed [CdCl₆₋ₙBrₙ]⁴⁻ would become a distorted octahedron due to the distinct bond lengths of Cd–Cl (2.66 Å) and Cd–Br (2.71 Å)[47], which may promote STEs at room temperature by the lattice distortion in excited states[48–50]. All the PLE spectra of CsCdCl₃:x%Br show gradually red-shifting and broadening peaks beyond 300 nm (Fig. 2a and Supplementary Fig. 9c–d), aligning well with the absorption spectrum (Supplementary Fig. 10a). This further suggests that transitions involving [e + a₁] →a₁ and [e + a₁] →e, a₁ in C₃ᵥ, as well as [eᵤ + a₂ᵤ] →a₁g in D₃d are all activated. Therefore, CsCdCl₃:x%Br samples exhibit a significantly enhanced PLQY up to 84.47% without relying on rare-earth metals (Fig. 2i and Supplementary Fig. 11), signifying a resource-saving approach for high-efficiency luminescence.

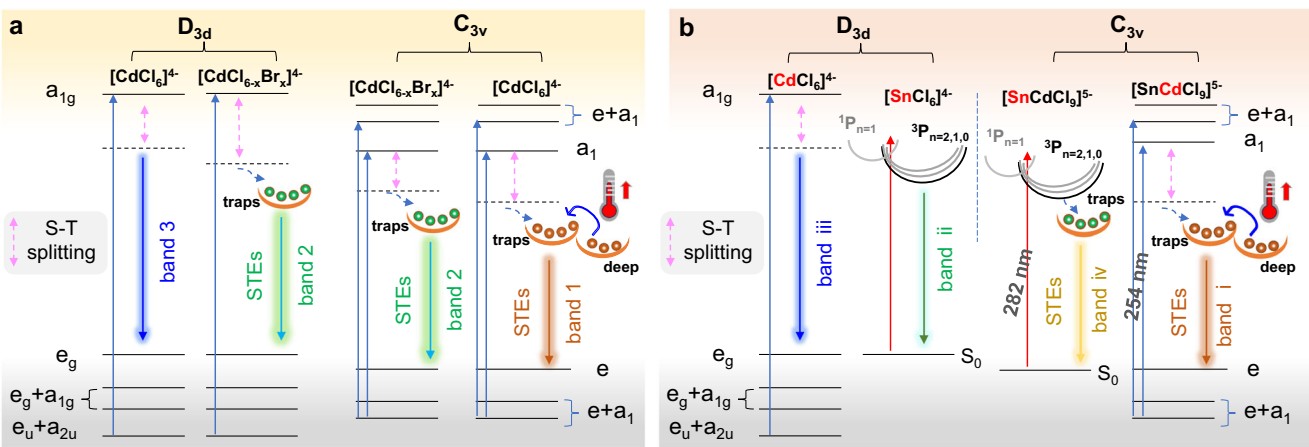

**Fig. 3 | The luminescent mechanism of Br- and Sn-doped perovskites.** PL mechanism and transition electronic energy level diagram of **a** Br⁻-doped and **b** Sn²⁺-doped CsCdCl₃ perovskites. S-T splitting: singlet–triplet-splitting. STEs: self-trapped excitons. D₃d and C₃v refer to D₃d and C₃v symmetry. Deep refers to deep traps. Thermometers imply that thermodynamics overcomes the energy barrier.

To understand the PL mechanism, temperature-dependent PL spectra for the representative of CsCdCl₃:0.8%Br and CsCdCl₃:10%Br were conducted. As illustrated in Supplementary Fig. 12a, the PL peak (band 3) emerges at low temperatures, followed by band 2 as temperature increases to room level, and then band 1 becomes more prominent at higher temperatures. Obviously, because of the low doping concentration of Br⁻ ions, both D₃d and C₃v exist in [CdCl₆]⁴⁻ and [CdCl₆₋ₙBrₙ]⁴⁻ forms, the band 3 assigned to pure [CdCl₆]⁴⁻ in D₃d symmetry at low temperature[45], and band 2 with broad emission corresponding to the Br-doped of [CdCl₆₋ₙBrₙ]⁴⁻ in both D₃d and C₃v symmetry, while band 1 only represents the undoped [CdCl₆]⁴⁻ in C₃v symmetry[48,49]. For the delayed spectra (Supplementary Fig. 12b), bands 3 and 2 survived owing to long-lived triplet exciton in both D₃d and C₃v symmetry. Upon 10% Br⁻ ions doping, the prompt spectra show that band 3 blends into band 2 at low temperatures due to the plentiful transformation of [CdCl₆]⁴⁻ into [CdCl₆₋ₙBrₙ]⁴⁻ in D₃d (Supplementary Fig. 12c), again confirming the band 3 originates from D₃d symmetry. In Supplementary Fig. 13, the Huang-Rhys factor (S) for 482 nm and 595 nm have been enhanced from 45 to 66 and 12 to 19 by increasing the doping concentrations of Br⁻ ions from 0.8% to 10%, respectively. This changed value of S indicates that increasing the doping concentrations of Br⁻ ions can lead to greater lattice distortion in D₃d and C₃v symmetry and enhance photon-phonon coupling, then forming different optical properties from pure CsCdCl₃ perovskites. Furthermore, such large S and significant Stokes shift, provide direct evidence that the LPL of band 2 and band 1 are associated with STEs[47,50]. The corresponding luminescence mechanism is depicted in Fig. 3a. It is noteworthy that band 1 becomes stronger with rising temperature to 377 K (Supplementary Fig. 12), illustrating its anti-thermal quenching ability, which will be further discussed below. Intriguingly, both the prompt and delayed spectra of CsCdCl₃:0.8%Br and 10%Br exhibit excellent temperature-dependent luminescent properties (Fig. 2e, f and Supplementary Fig. 14). Of that, CsCdCl₃:0.8%Br displays remarkable color variation, ranging from blue to cyan, then across yellow-green and finally to orange–red, observable with naked eyes (Fig. 4d), in good agreement with CIE coordination (Fig. 4b, c). Such a wide range of full-color-tunable luminescence and the anti-thermal quenching properties are still rare, particularly in state-of-the-art LPL materials (Supplementary Table 5).

Remarkably, CsCdCl₃:x%Br samples exhibit substantial LPL with distinctive time-dependent afterglow alterations. As illustrated in Fig. 4a, the pristine CsCdCl₃ host manifests an orange–red afterglow extinguishing promptly upon cessation of the 254 nm UV lamp. CsCdCl₃:x%Br, conversely, hold a bright blue-green color when exposed to 254 nm UV light. Subsequent to excitation cessation, CsCdCl₃:x%Br exhibit color-variable LPL from blue-green to orange–red, effectively covering the entire visible spectrum. Notably, unlike conventional time-dependent luminescence, herein the time–valve in the color change can be regulated based on Br⁻ ion concentration. To elucidate this phenomenon, steady-state luminescent decay curves were monitored at emission centers of 482 and 595 nm. As shown in Fig. 2g, the afterglow intensity at 595 nm rapidly diminishes in the first 500 s, followed by a gradual decline extending up to 2000 s to discern from the background. The decay lifetime of 482 nm in the initial 60 s window is extended with increasing the Br⁻ ion concentration to 10% (Fig. 2h). As observed in Supplementary Movie 1, the afterglow persists for 1800 s to naked eyes. Time-dependent emission spectra and the corresponding CIE of CsCdCl₃:0.2 ~ 0.8%Br also show that the color can be changed by the decay time (Supplementary Fig. 15). Analysis of time-resolved PL mapping in Fig. 2b and Supplementary Fig. 16 indicate that the LPL of 595 nm remains consistently strong, while LPL of 482 nm can be regulated for different concentration of Br⁻ ions. The time-dependent afterglow spectrum vividly demonstrates that the LPL intensity at 595 and 482 nm can be tuned by Br⁻ ion doping concentrations, and then changed by time evolution due to different decay lifetimes. (Fig. 2c and Supplementary Fig. 17). Hence, the color-variable LPL can be interpreted as the synergistic concurrent of different decay lifetimes at 482 nm and 595 nm, which are modulated by varying concentrations of Br⁻ ions in CsCdCl₃:x%Br.

Charge trap state analysis through TL measurements is an effective method for elucidating LPL. Firstly, CsCdCl₃:x%Br show good thermal stability (Supplementary Fig. 18a). No TL signal is detected for the pristine CsCdCl₃, implying the absence of LPL nature (Fig. 2d). For CsCdCl₃:x%Br, four different cases arise: (i) x = 0.2 ~ 0.5, the trap energy level (E_trap) was mainly determined at 0.74 eV; (ii) x = 0.8 ~ 5, these E_trap were distributed around at 0.73, 0.78 ~ 0.93 eV; (iii) x = 10 ~ 15, the E_trap occured at 0.71 ~ 0.74 eV, where the E_trap can be estimated by Urbach's empirical formula E_trap = T_m/500 (T_m is the temperature of TL peak)[51]. These CsCdCl₃:x%Br samples all possess shallow traps ranging from 0.67 to 0.76 eV, preferable for creating an ideal depth for LPL[52]. In addition, the trap depths in the range of 0.8–1.6 eV are categorized as deep traps, typically resulting in low LPL at room temperature due to the activation energy barrier[53]. However, as the temperature increases, the charge carriers in the deep traps overcome the activation energy barrier and migrate to the emission center, leading to an anti-thermal quenching property.

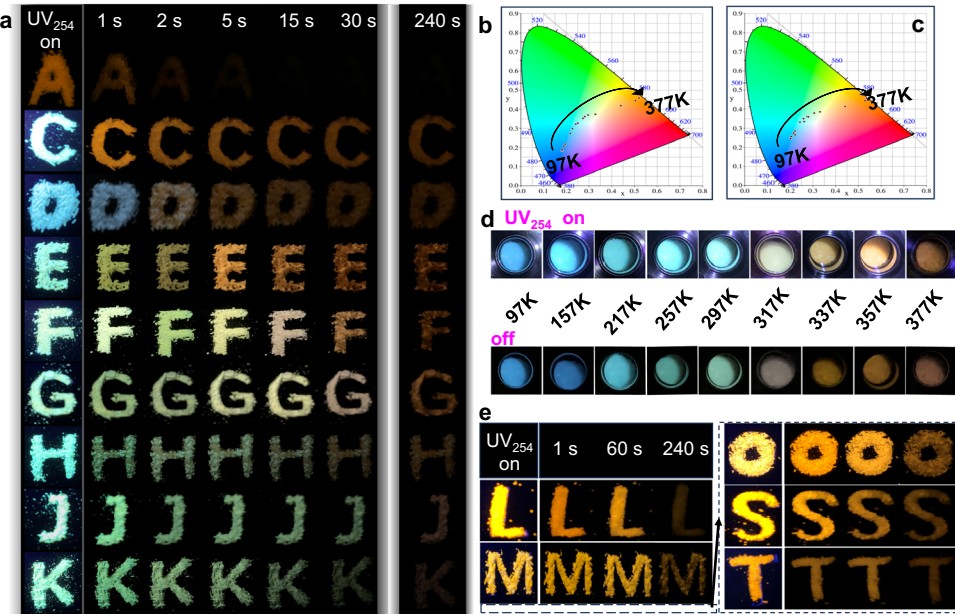

**Fig. 4 | Luminescent and afterglow behaviors of CsCdCl₃:x%Br and CsCdCl₃:x% Sn. a** Afterglow photographs for CsCdCl₃:x%Br, A (x = 0), C (x = 0.2%), D (x = 0.5%), E (x = 0.8), F (x = 1), G (x = 3), H (x = 5), J (x = 10) and K (x = 15). CIE coordinate diagram of CsCdCl₃:0.8%Br in temperature-responsive **b** prompt and **c** delayed mode. **d** Temperature-dependent PL emission color of CsCdCl₃:0.8%Br were controlled by switching on/off the 254 nm UV lamp. **e** Afterglow photographs for CsCdCl₃:x%Sn, L (x = 1), M (x = 3), O (x = 5), S (x = 10) and T (x = 15).

To further prove this hypothesis and elucidate the phenomenon of color-changing afterglow, "thermal cleaning" method was conducted for CsCdCl₃:0.8%Br and CsCdCl₃:10%Br. As shown in Supplementary Figs. 19a, 20a the TL intensity decreased while $T_m$ increased with rising excitation temperature, which can be attributed to the redistribution of charge carriers from deep traps to shallow traps (STs)[54]. The $E_{trap}$ and excitation temperature exhibit a strong linear relationship (Supplementary Figs. 19b, 20b), which directly indicates the presence of a quasi-continuous distribution of trap depths in the shallow region[55]. The initial-rise method was employed to analyze the shallowest trap levels, wherein the Arrhenius equation was utilized to ascertain that the concentration of trap-carriers remains relatively constant on the low-temperature side of the TL curve (Supplementary Fig. 19c, 20c). It is worth noting that the shallowest trap levels do not converge to the average values of 0.17 eV and 0.41 eV (Supplementary Figs. 19d, 20d), respectively. These results strongly indicate that afterglow activity levels are non-narrowly distributed and beneficial to the formation of the color-changing afterglow[56]. The shallow trap depths of CsCdCl₃:0.8%Br and CsCdCl₃:10%Br were estimated to be 0.53 eV and 0.73 eV (Supplementary Figs. 19e, 20e), respectively, based on the Hoogenstraaten method[57], providing further evidence that the free storage and release of charge carriers from the traps is an important factor in ensuring ultralong-lived emission. Furthermore, the 3D TL spectra are used to analyze color-changing afterglow based on the relationship between the emission and TL spectrum. As shown in Supplementary Figs. 21c, d and 22c, d, the emission centers at 595 nm are concentrated at 319 K and 421 K in CsCdCl₃:0.8%Br, and at 327 K and 416 K in CsCdCl₃:10%Br, implying the LPL and anti-thermal quenching properties of 595 nm are empowered by both deep and shallow traps. The emission centers at 482 nm initially manifest at a temperature of 295 K (0.59 eV) in CsCdCl₃:0.8%Br (Supplementary Fig. 21b), and then can be sustained as the temperature rises to 327 K (0.65 eV) in CsCdCl₃:10%Br (Supplementary Fig. 22b, c). These results demonstrate that by modulating concentration of Br⁻ ions, the trap at 482 nm can be adjusted closer to the ideal trap depth (0.67 ~ 0.76 eV) for LPL, resulting in a longer afterglow lifetime at 482 nm, which is highly consistent with the luminescent decay curves at 482 nm

(Fig. 2h). Therefore, the controllable color-varying long-afterglow can ultimately be ascribed to the tunable trap states that arise from variations in Br⁻ ion concentration within the broken symmetry skeleton. These new forming traps can influence the decay lifetimes at 482 nm and 595 nm. Upon cessation of excitation, charge carriers stored at the tunable traps can be released and radiated at 482 nm with more variable decay times and intensities, subsequently cooperating with the LPL of 595 nm spring from the ideal trap depth, thereby giving rise to this remarkable phenomenon. The associated mechanism is shown in Fig. 3a.

The same host material exhibits a variety of luminous capabilities, offering the potential for multifunctional applications. In the case of controlled color-changing afterglow achieved by Br⁻ ions doping engineering, we continue to extend the luminescence functionality of the CsCdCl₃ host and choose the 5s² electronic configuration of Sn²⁺ ions as activators. As illustrated in Fig. 5a and Supplementary Fig. 23b–d, under the excitation of 254 nm UV light, a predominant emission peak at 595 nm is significantly enhanced in both prompt and delayed spectra with increasing Sn²⁺ ion dopant concentration from 1 to 10%. However, further doping results in a slightly reduced peak intensity. CsCdCl₃:10%Sn displays the highest PLQY up to 65.71% (Fig. 5f and Supplementary Fig. 24). Subsequently, the PLQY experiences a marginal decline with a further increase of Sn²⁺ ion concentration, attributed to intensified Sn²⁺–Sn²⁺ dipole interactions causing nonradiative energy transition. The prompt and delayed PLE spectra exhibit a primary peak centered at 254 nm (Supplementary Fig. 23a, c), with an apparent shoulder peak at 282 nm becoming more pronounced as Sn²⁺ ion concentration increasing (Supplementary Fig. 23e, g), consistent well with the absorption spectra (Supplementary Fig. 10b).

As depicted in Fig. 5a and Supplementary Fig. 23f, h, an obvious emission peak at 565 nm is enhanced upon increasing the Sn²⁺ ions under the excitation wavelength of 282 nm. To further investigate this uncommon phenomenon, a series of excitation wavelength-dependent PL spectra and 2D excitation maps were performed for these CsCdCl₃:x% Sn. As shown in Fig. 5i and Supplementary Figs. 25–29, as the excitation wavelength red-shifts from 250 to

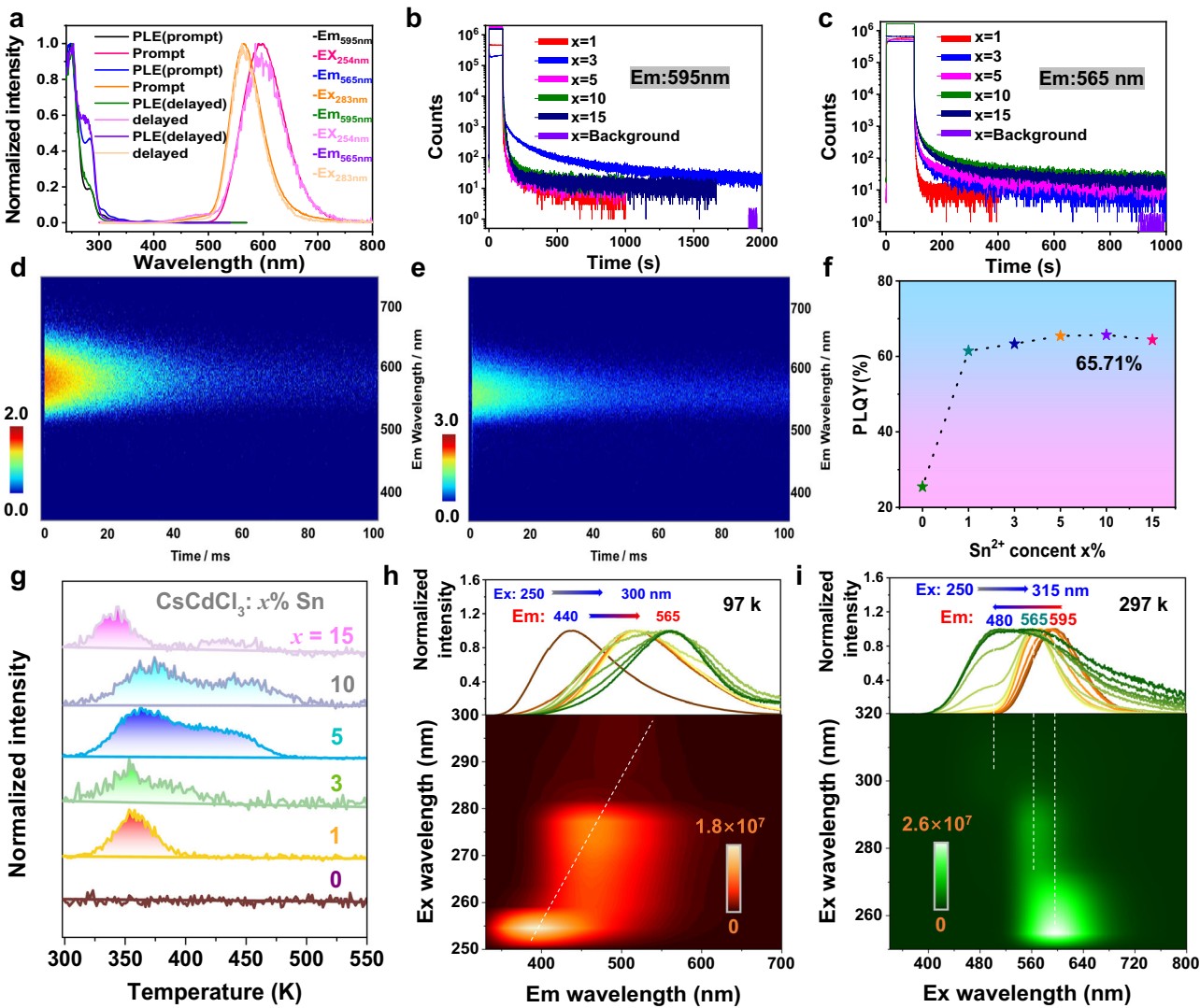

**Fig. 5 | Luminescent performances of CsCdCl$_3$:$x$%Sn. a** Normalized PL spectra of CsCdCl$_3$:10%Sn based on prompt and delayed (t$_d$ = 1 ms) patterns under 254 nm or 282 nm excitation, along with the corresponding normalized PLE spectra. Afterglow decay curve of CsCdCl$_3$:$x$%Sn with the detected emission wavelength **b** 595 nm and **c** 565 nm. Pseudo color map of time-resolved PL spectra of afterglow for CsCdCl$_3$:10%Sn under **d** 254 nm and **e** 282 nm flash lamp. **f** The PLQY of CsCdCl$_3$:$x$%Sn. **g** TL spectra of CsCdCl$_3$:$x$%Sn. Excitation-prompt mapping of CsCdCl$_3$:10%Sn under **h** 97 K and (**i**) 297 K. Note: Due to limited space of the picture, only the excitation wavelengths of the start and end are shown. The color gradient of curves in Fig. 4h from left to right and in Fig. 4i from right to left solely represents the change in emission peaks based on the relevant excitation wavelength (ex: 250, 255, 260, 265, 270, 275, 280, 285, 290, 295, 300, 305, 310 and 315 nm, respectively), without any additional connotations.

285 nm, the emission band center at 595 nm undergoes a significant blue shift to 565 nm with reduced intensity in both prompt and delayed spectra. Continuous shifting of the excitation to 315 nm results in a faint peak at 480 nm, which is negligible in the 2D excitation map (Fig. 5i and Supplementary Figs. 25–29). Notably, such a large contrast in the blue-shifted emission based on continuous red-shifted excitation at room temperature has not been reported in halide perovskites, even rarely among various current luminous materials (Supplementary Table 5).

To elucidate this intriguing phenomenon, we conducted temperature-dependent PL spectra for representative CsCdCl$_3$:3%Sn and CsCdCl$_3$:10%Sn. As shown in Supplementary Figs. 30, 31, under 254 nm irradiation at low temperatures, the initial observation is the emergence of prompt PL band iii, faintly mirrored in the delayed spectrum, alongside the prompt PLE peak at 254 nm (Supplementary Fig. 32). These findings suggest that the band iii originates from unaltered [CdCl$_6$]$^{4-}$ in D$_{3d}$ symmetry. As the temperature rises to 177 K, a broad prompt PL band ii becomes prominent, replacing the

weakened band iii, a change similarly in the delayed spectra, indicating their common origin in triplet excitons (Supplementary Figs. 30b, 31b). Considering Sn$^{2+}$ ions with a 5s$^2$ electron configuration and $^1$P$_1$/$^3$P$_{0,1,2}$ energy levels[58], transitions such as $^1$S$_0$ → $^3$P$_0$ and $^1$S$_0$ → $^3$P$_2$ are deemed forbidden, while allowed transitions include $^1$S$_0$ → $^3$P$_1$ and $^1$S$_0$ → $^1$P$_1$ due to spin-orbit coupling[59,60]. The broad emission band ii is primarily attributed to the $^3$P$_1$ → $^1$S$_0$ transition of Sn$^{2+}$ ions situated in D$_{3d}$ [SnCl$_6$]$^{4-}$ symmetry, partially involving the breaking of the forbidden transition $^3$P$_2$ → $^1$S$_0$ at low temperatures[61]. The Huang-Rhys factor (S) at 595 nm has been determined to range from 16 to 25 (Supplementary Fig. 33a, b). These results directly demonstrate that band i (595 nm) is also associated with STEs due to the elevated proportion of Cd$^{2+}$-based octahedral distortion with [SnCdCl$_9$]$^{5-}$ moieties.

To further substantiate this hypothesis, we adjusted the excitation wavelength to 282 nm. As illustrated in Supplementary Figs. 34–35, the PL band iii diminishes, while band ii remains robust at low temperatures, affirming their origin from [CdCl$_6$]$^{4-}$ and [SnCl$_6$]$^{4-}$ in D$_{3d}$ symmetry, respectively. With increasing temperature, a newly emerging

band iv with an emission center at 565 nm gains strength progressively. This observation is influenced by two main factors: (a) The lower excitation energy of 282 nm can effectively populate the charge carriers from the $5s^2$ orbital of $Sn^{2+}$ into the lower-energy $^3P_1$ excited state, and further be saved by traps. These charge carriers also can escape from shallow traps or deep traps by thermodynamically breaking the energy barrier to forming the anti-thermal quenching ability[59,60,62]. (b) The charge carriers migrate back to the luminescent center and undergo electron-phonon coupling, resulting in the formation of LPL and STE of 565 nm. (c) Such high Huang-Rhys factor (S) (Supplementary Fig. 33a, b) and large Stokes shift crucially support the STEs of 565 nm[63]. Further, a series of excitation-dependent PL experiments were conducted on $CsCdCl_3$:3%Sn and $CsCdCl_3$:10%Sn at 97 K. As shown in Fig. 5h and Supplementary Figs. 36a–37a, varying the excitation wavelength from 250 to 300 nm results in a red-shift of the prompt emission center from 440 to 565 nm, corresponding well with the above-mentioned band iii, band ii and band iv. In the delayed spectra (Supplementary Figs. 36b–37b), the emission band at 595 nm (band i) decreases, and the broad emission band centered at 525 nm (band ii) intensifies, confirming that optical properties are influenced by the geometric symmetry and energy states of the metal centers[62–64]. The corresponding luminescence mechanism is depicted in Fig. 3b. The distinctive feature as Janus-type luminescence—forward excitation-dependence at low temperature and the reverse excitation-dependence at room temperature—has not been reported previously (Supplementary Table 5), holding promise for potential applications in information safety and temperature recognition. Interestingly, this kind of materials can maintain high luminous intensity in ultra-pure water and acid (Supplementary Fig. 38). Furthermore, after 46 days of storage at room temperature, all samples exhibit no significant spectral changes (Supplementary Figs. 39, 40), underscoring the excellent optical stability of $CsCdCl_3$:x% Sn.

Remarkably, all $CsCdCl_3$:x%Sn samples emit orange–red LPL after the cessation of 254 nm UV light (Fig. 4e and Supplementary Movie 1). The steady-state luminescent decay of the emission bands at 595 nm and 565 nm were investigated. Following the termination of 254 nm irradiation, the intensity of the $CsCdCl_3$: x %Sn at 595 nm rapidly decreases in an initial 500 s and persists for up to 2000 s (Fig. 5b). For Fig. 5c, the intensity of the $CsCdCl_3$:10%Sn at 565 nm also decays quickly in the first 400 s, and gradually slows down until reaching 1000 seconds. The characteristics of LPL at 595 nm and 565 nm are demonstrated through time-resolved PL mapping (Fig. 5d, e and Supplementary Fig. 41) and the time-dependent afterglow emission spectrum (Supplementary Fig. 42). Moreover, $CsCdCl_3$:x%Sn exhibit good thermal stability (Supplementary Fig. 18b). Focusing on the TL spectra (Fig. 5g), $CsCdCl_3$:x%Sn exhibit distinct trap energy levels (estimated by $E_{trap} = T_m/500$) with different x values, where they all display shallow trap energy levels around 0.68 - 0.71 eV, as well as deep trap energy levels ranging from 0.79 to 0.90 eV with x = 3 - 15. It is worth noting that the TL properties of $CsCdCl_3$:x%Sn are similarly to those of $CsCdCl_3$:x%Br, attributed to the formation of trap states caused by similar Cl, Cs, and Cd ions vacancies or lattice dislocations in the broken $CsCdCl_3$ skeleton during $Sn^{2+}$ or $Br^-$ ion doping. Given this, it is no surprise that traps play a crucial role in saving charge carriers from the excited state, evoking the LPL of these $CsCdCl_3$:x%Sn by releasing charge carriers from shallow traps or thermally breaking the energy barrier for migration from deep traps, and then to the ground state. To further verify this assumption, we also employ the "thermal cleaning" method to investigate the TL of $CsCdCl_3$:10%Sn. As illustrated in Supplementary Fig. 43a, b, the excitation temperature increment leads to the redistribution of charge carriers by facilitating their transfer from deep to shallow traps. Furthermore, the well-established correlations between trap energy and excitation temperature elucidate the quasi-continuous distribution of trap depths.

These shallowest trap levels do not converge to an average value of 0.14 eV (Supplementary Fig. 43c,d), indicating the presence of multiple afterglow activity levels that provides the possibility of multiple LPL in this system, herein supporting the LPL of 565 and 595 nm. The shallow trap depths of $CsCdCl_3$:10%Sn can be determined to be 0.51 eV (Supplementary Fig. 43e). The 3D TL spectra exhibit three TL centers at 353 K, 390 K and 423 K (Supplementary Fig. 44), respectively. Obviously, the wavelength-resolved TL peak at 353 K (0.71 eV) is broader than that at 390 K (0.78 eV) and 423 K (0.85 eV), and it can be well fitted as two peaks by a Gaussian function. This indicates that both emissions at 565 and 595 nm can be caused by an ideal shallow trap, leading to the formation of LPL. Hence, these results are in good agreement with the experimental phenomena and the proposed mechanism (Fig. 3b).

## Density Functional Theory (DFT)

To gain deeper insights into the electronic structure and color-tunable luminescence mechanisms by disrupting the original symmetry and the doping effect in the all-inorganic skeleton, DFT calculations were performed on ten idealized models of $CsCdCl_3$ doped with Br and Sn (Supplementary Fig. 45). Their lattice constants (V) exhibit an enlargement to further confirm the expansion of the host lattice in XRD patterns (Supplementary Table 6, Supplementary Figs. 4, 5). We conducted band structure and total density of states (DOS) calculations. The pristine $CsCdCl_3$ exhibits a direct bandgap, whereas the Br-doped and Sn-doped models exhibit indirect bandgaps (Supplementary Figs. 46, 47), thereby mitigating hole-electron recombination and extending exciton lifetimes[65,66]. In Fig. 6a and Supplementary Figs. 48–53, the projected DOS (PDOS) of $D_{3d}$-$Br_i$-$C_{3v}$ (i = 1,2,3) and $C_{3v}$-$Br_j$-$C_{3v}$ (j = 4,5,6) models reveal that the valence band (VB) is mainly composed of Cl $3p$ and Br $4p$ orbitals, while the conduction band (CB) consists of Cd $5s/5p$, Br $4p$, and Cl $3p$ orbitals. In contrast, Cs orbitals play a negligible role in the band structures of these models. The charge density maps illustrate that the valance band maximum (VBM) is localized at $Br^-$ and $Cl^-$ ions in $[CdCl_{6-n}Br_n]^{4-}$ moiety of both $D_{3d}$ and $C_{3v}$ symmetry, while the conduction band maximum (CBM) is predominantly formed from $Cd^{2+}$ (Fig. 6c, Supplementary Figs. 49b–53b). The small discrepancy in bandgaps between the $D_{3d}$-$Br_i$-$C_{3v}$ (i = 1,2,3) and $C_{3v}$-$Br_j$-$C_{3v}$ (j = 4,5,6) models, approximately 12.2 meV, has negligible impact on the PL of $[CdCl_{6-n}Br_n]^{4-}$ in both $D_{3d}$ and $C_{3v}$ symmetries (Supplementary Fig. 46g). This reaffirms that the broad emission band at 482 nm (band 2) in Br-doped $CsCdCl_3$ originates from the combined emission center of $[CdCl_{6-n}Br_n]^{4-}$ in both $D_{3d}$ and $C_{3v}$ symmetries (Figs. 2a, 3a and Supplementary Fig. 9).

For Sn-doped $CsCdCl_3$ (Fig. 6b, Supplementary Figs. 54–56), PDOS shows that VBs consist of Cl $3p$ and Sn $5s$ orbitals, while the CBs comprise Cl $3p$, Sn $5p$ and Cd $5s/5p$ orbitals in $D_{3d}$-$Sn_i$ (i = 1,4) and $C_{3v}$-$Sn_j$ (j = 2,3) models. The bandgap of $D_{3d}$-$Sn_i$ (i = 1,4) models is larger than $C_{3v}$-$Sn_j$ (j = 2,3) models, around 30.4–95.7 meV (Supplementary Fig. 47f), impacting the PL characteristics of $[SnCl_6]^{4-}$ in both $D_{3d}$ and $C_{3v}$ symmetries due to exceeding the thermal energy of 26 meV[67]. This observation aligns with the experimental trend of Sn-doped materials exhibiting a reduced bandgap (Supplementary Fig. 10d). Additionally, these results underscore that the luminescence center of $Sn^{2+}$ ions in $C_{3v}$ $[SnCdCl_9]^{5-}$ symmetry requires less matching excitation energy, resulting in the reverse excitation-dependent behavior observed in experiment. From Fig. 6d and Supplementary Figs. 54–56b, the charge density maps highlight $Sn^{2+}$ ion with unique 5s-related energy levels significantly influencing the VBM and contributing to multimode luminescence. Interestingly, when $Sn^{2+}$ ion doping at $C_{3v}$ $[SnCdCl_9]^{5-}$ symmetry (Fig. 6d and Supplementary Figs. 54b–56b), the charge density of CBM is primarily located at the $Cd^{2+}$ ion of $C_{3v}$ symmetry, further supporting the hypothesis that band i stems from $C_{3v}$ symmetry (Figs. 5a, 3b and Supplementary Figs. 30, 31).

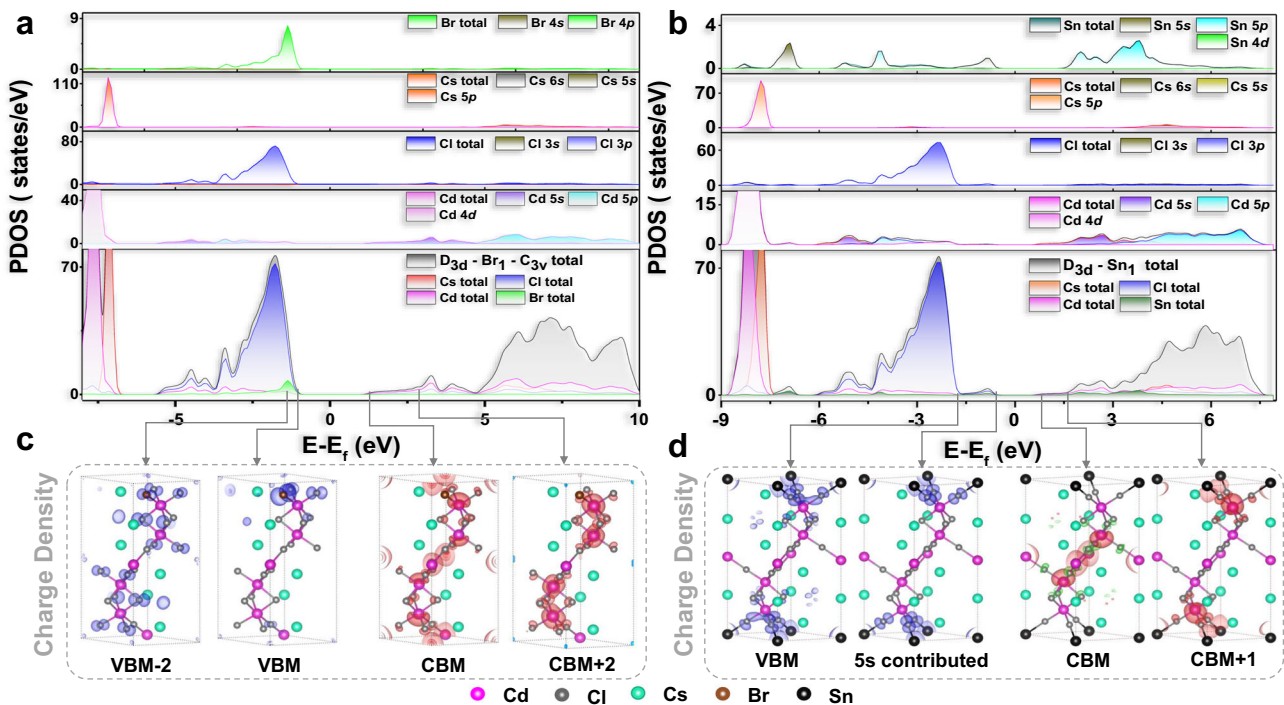

**Fig. 6 | DFT calculations of Br- and Sn-doped perovskites.** PDOS of **a** $D_{3d}$-$Br_1$-$C_{3v}$ and **b** $D_{3d}$-$Sn_1$ model. Visualization of the Gamma point with VBM and CBM-associated charge density maps in **c** $D_{3d}$-$Br_1$-$C_{3v}$ and **d** $D_{3d}$-$Sn_1$ model, as well as H point for 5s-contributed charge density map. PDOS: projected density of states.

## High-level anti-counterfeiting using multifunctional LPL

Programming advanced anti-counterfeiting technology is of considerable practical importance, by leveraging the LPL characteristics of all-inorganic halide perovskites to demonstrate their distinctive spatial-time-resolved, and time-logical color-variable afterglow properties. As depicted in Supplementary Fig. 57, no perceptible alterations are observed under ambient lighting conditions. However, upon exposure to 254 nm UV irradiation (Fig. 7a), the combination of chaotic orange–red and cyan colors could potentially convey misleading messages such as "I • BNU 8888" (BNU: Beijing Normal University) and two other false messages when individually viewed in the orange–red or cyan channel. After ceasing the UV lamp for 1 s (Fig. 7b), the subsequent error information is interpreted as "I • BNU 2083", accompanied by two additional error messages, resembling a three-dimensional (3D) encryption featuring spatial-time-dual-resolved patterns. The cyan "•" transfers to milk white, and some local areas exhibit color changes in 3 s (Fig. 7c). Despite the message still reading as "I • BNU 2083", it has evolved into the 4D anti-counterfeiting category due to its variable color factor. Ultimately, a cohesive orange-yellow color palette effectively conveys the intended message of "I • BNU 2023" (Fig. 7d and Supplementary Movie 2). The overall process can be regarded as a 5D anti-counterfeiting technology, which is anticipated to surpass conventional afterglow materials due to its additional time-gated color change facilitated by Br-doping engineering.

Furthermore, proof-of-concept experiments were conducted by filling a series of as-synthesized perovskites into the QR code groove (Supplementary Fig. 58). As depicted in Fig. 7e, the security model operates under a mechanism wherein specific attributes are assigned to the QR code of each region: emission loss (marked in red), QR code with afterglow duration every 0.5 s (marked in blue), and modification of the afterglow color (marked in green). These attributes correspond to the binary shifts of one bit (+1) in the respective regions. Upon excitation by the 254 nm UV lamp, the QR code displays chaotic orange–red and cyan hues, representing the binary exchange algorithm "000". At a delay time of 0.5 s, region C undergoes the first

transition from cyanogen to cyanogen yellow, introducing binary ciphers for "011", "110" and "101". One second later, the emission from region A is nearly extinguished, while region C transforms to orange-yellow, resulting in adjustments of the binary ciphers to "11010", "10110" and "10101". After 18 s, the new binary ciphers take form, generating security codes "1161, 1764, 1508, 1481, 1175 and 1179" through binary and decimal conversion (Fig. 7h). According to algorithm that we have set with "Time code (at 18 s) - QR code (at 18 s) - color code (at 18 s)", the final lock code is determined as "1508" (Supplementary Fig. 59a). It is noteworthy that the above algorithm is applicable only when i = j = n at the respective time nodes. If all codes could be exchanged (Fig. 7f), it would result in an information Big-Bang, achieving the maximal information loading capacity (Fig. 7f–h, Supplementary Fig. 59b). Therefore, these perovskites are anticipated to be highly effective for advanced high-security anti-counterfeiting due to leveraging the advantages of time-sensitive color and spatial-time four-resolved functionality.

## Discussion

In summary, we have successfully demonstrated an ultralong (> 2000 s) persistent luminescence by incorporating $Br^-$ or $Sn^{2+}$ ions into the hexagonal phase $CsCdCl_3$, achieving the highest recorded PLQY (84.47%) among current halide perovskites. The simultaneous significant improvement in afterglow lifetime and efficiency can be attributed to the reconstruction of the luminescence center induced by doping, leading to a disruption of the local symmetry in the host framework, as well as the formation of tunable traps. Unlike conventional long-afterglow materials, $CsCdCl_3$:$x$%Br exhibits a precisely regulatable color change time valve, determined by varying $Br^-$ ion doping concentrations, enabling both time- and temperature-dependent LPL. The unique $5s^2$ electron configuration of $Sn^{2+}$ ions, coupled with distinct geometric symmetries that construct multiple luminescence centers, results in forward and reverse excitation-dependent PL behavior of $CsCdCl_3$:$x$%Sn at low and room temperatures, respectively. Therefore, this work not only addresses the existing gap in the field concerning wide-range full-color long-afterglow

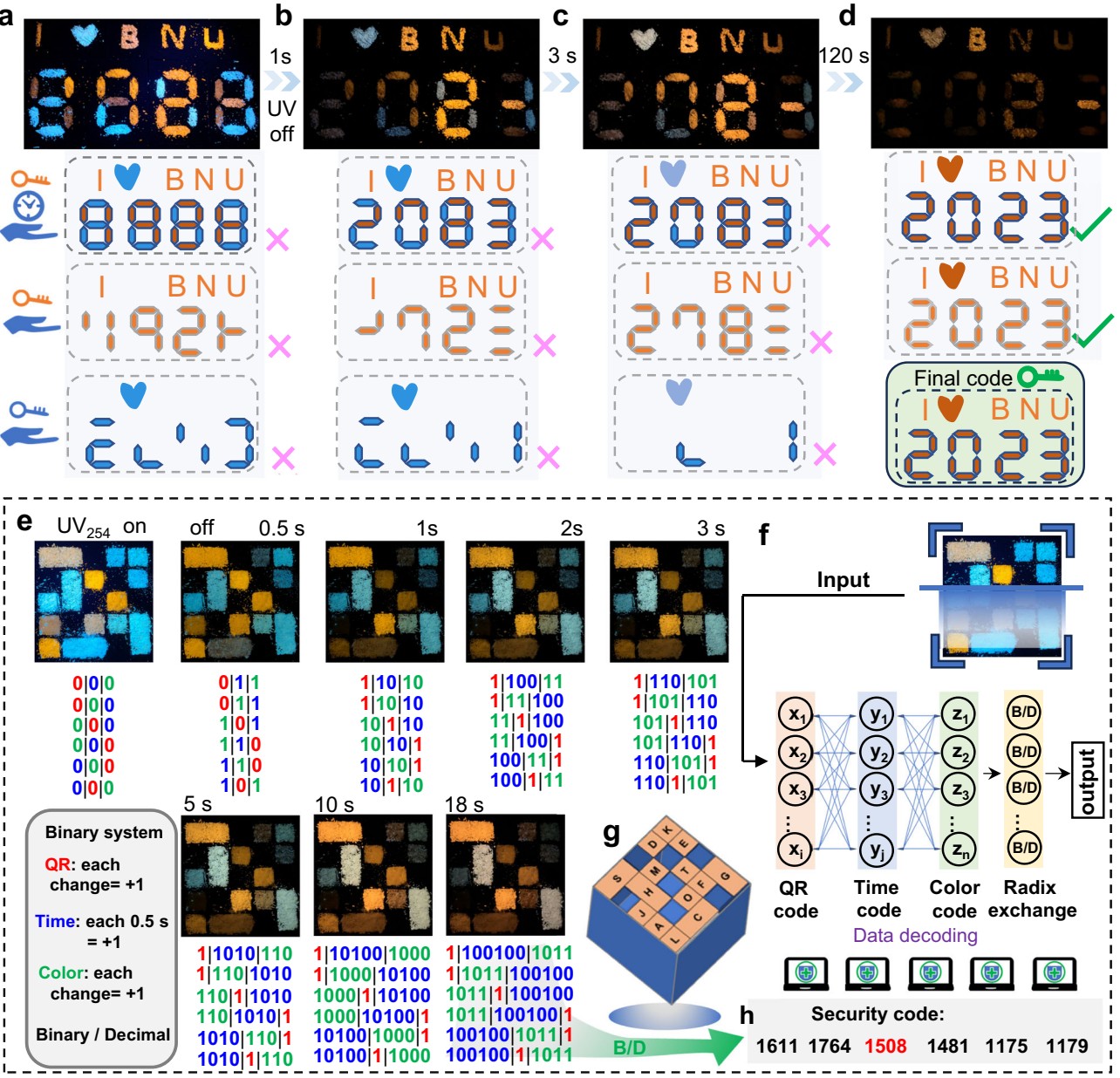

**Fig. 7 | Time–valve controlled color for multilevel security. a** Photographs of labels prepared by CsCdCl$_3$:$x$%Br and CsCdCl$_3$:$x$%Sn sample powders under 254 nm irradiation, along with corresponding afterglow emission after ceasing excitation at **b** 1 s, **c** 3 s and **d** 120 s. **e** QR code image of CsCdCl$_3$:$x$%Br and CsCdCl$_3$:$x$%Sn under 254 nm UV lamp, followed by afterglow imaging at different time intervals. **f** Schematic illustration of the QR code of the encryption and decryption process. **g** QR code map including CsCdCl$_3$:$x$%Br and CsCdCl$_3$:$x$%Sn, with A (0%Br), C(0.2% Br), D (0.5%Br), E (0.8%Br), F (1%Br), G (3%Br), H (5%Br), J (10%Br), H (15%Br), L (1% Sn), M (3%Sn), O (5%Sn), S (10%Sn) and T (15%Sn). **h** Conversion between binary and decimal for QR code with afterglow imaging after 18 s.

and Janus-type Ex-De materials but also introduces a significant paradigm for multifunctional LPL, with applications in high-security anti-counterfeiting and 5D information coding and storage.

## Methods
### Materials
Cesium chloride (CsCl, 99.99%, Innochem), Cadmium chloride (CdCl$_2$, 99.99%, Aladdin), Cadmium Bromide (CdBr$_2$, 99.99%, Adamas), Cesium Bromide (CsBr, 99.99%, Adamas), Tin(II) chloride (SnCl$_2$, anhydrous, 99.99%, Aladdin), Hydrobromic acid (HBr, AR, 40 wt.% solution in water, Macklin), H$_3$PO$_2$ (50 wt.% solution in water, Sigma Aldrich). Hydrochloric acid (HCl, 12 M) was purchased from Xilong Scientific Co. Ltd.

### Preparation of Br-doped CsCdCl$_3$ single crystals
For pure CsCdCl$_3$ single crystals (SCs), 4 mmol of CsCl, 4 mmol CdCl$_2$ were dissolved in 20 mL 12 M hydrochloric acid. The solution was heated at 180 °C for 12 h in a stainless steel autoclave reactor and then was programmed cooling to room temperature (RT, referring to 25 °C) at a speed of 5 °C/h, which is different from the traditional hydrothermal method by natural cooling. The crystals at the bottom were rinsed with isopropanol before drying on a filter paper. For CsCdCl$_3$:$x$% Br crystals, 4×(1−$x$) mmol of CsCl, 4×(1−$x$) mmol CdCl$_2$, 4×$x$ mmol of CsBr and 4×$x$ mmol of CdBr$_2$ were dissolved in a total 20 mL mix solution of hydrochloric acid and hydrobromic acid with molar ratio of (1−$x$)/$x$. the other procedures are the same as above. All crystals were preserved in a caped vial for further characterization.

## Preparation of Sn-doped CsCdCl₃ single crystals

According to the above method, replacing raw materials with 4 mmol of CsCl, 4×(1−x) mmol CdCl$_2$ and 4×x mmol SnCl$_2$ to dissolve in 20 mL 12 M hydrochloric acid with 50 wt.% H$_3$PO$_2$ (~132 μL, 1 mmol). Finally, CsCdCl$_3$:x%Sn crystals were harvested.

## Characterizations

Single-crystal X-ray diffraction data of these samples were investigated by Rigaku Oxford Diffraction Supernova X-ray source diffractometer equipped with monochromatized Mo-Kα radiation ($\lambda = 0.71073$ Å) at 100 K. Scanning electron microscope (SEM) were characterized by Hitachi SU8010 instrument, and the corresponding element content was collected by energy disperse spectrometer (EDS, Oxford X-Max Aztec). X-ray Photoelectron Spectroscopy (XPS) data were collected using EscaLab 250Xi instrument. Solid UV–vis absorption spectra were collected on a Shimadzu UV-3600 spectrophotometer at room temperature with the wavelength range of 240–500 nm, and BaSO$_4$ powder was used as a standard (100% reflectance). TGA tests were collected on a Perkin-Elmer Diamond SII thermal analyzer under the atmosphere of nitrogen with a heating rate of 10 K min$^{-1}$. All the relevant photoluminescence (PL) tests and time-resolved lifetime were conducted on an Edinburgh FLS980 fluorescence spectrometer. The PLQY values were acquired using a Hamamatsu Quantaurus-QY Spectrometer (Model C11347-11) equipped with a xenon lamp, integrated sphere sample chamber and CCD detector. The TL spectra were determined by Risφ TL/OSL Da-20 (DTU Nutech, Denmark) instrument with samples were pre-irradiated under 254 nm UV lamp for 1 min at RT. The 3D TL spectra were determined by TOSL-3DS. The ICP-OES was performed on Agilent 720ES. The software Origin 2021 and Microsoft PowerPoint are used for visualizing graphics and analyzing data. The fluorescence and bright field crystal images were captured using an OLYMPUS IXTI fluorescence microscope (Olympus Corporation, Tokyo, Japan). Additionally, all other visual images in this article were acquired utilizing an iPhone 12.

## Theoretical calculations

The calculations were performed with the density functional theory (DFT) by Quantum ESPRESSO (qe-7.2)[68]. The generalized gradient approximation of the Perdew–Burke–Ernzerhof (PBE) parameterization with projector-augmented wave method are performed for the exchange and correlation functional. Wavefunctions expanded in plane waves were cut off to 60 Ry kinetic energy. The computational models encompass all the potential sites by disrupting original symmetry in all-inorganic skeletons through doping with Br$^-$ or Sn$^{2+}$ ions. Such as CsCdCl$_3$:5.5%Br model was constructed by replacing one Cl atom with a Br atom in a 30-atom cell model, which is analogous to the CsCdCl$_3$:5%Br model. Similarly, the CsCdCl$_3$:16.6%Sn model, achieved by substituting one Cd atom with a Sn atom, which is analogous to the CsCdCl$_3$:15%Sn. The Brillouin zone was sampled using a 15×15×6 Monkhorst–Pack k-mesh, which was examined have good convergence. The Γ point was used to represent the Brillouin zone. For the elements Cs, Cd, Sn, Br and Cl, ultra-soft pseudopotentials are used. The energy convergence criterion is set as $1.0 \times 10^{-5}$ eV for structural relaxations. The lattice constants were optimized and compared with experimental results in Supplementary Table S6.

## Reporting summary

Further information on research design is available in the Nature Portfolio Reporting Summary linked to this article.

## Data availability

All data needed to evaluate the conclusions in the paper are present in the paper and/or the Supplementary Materials. The accession number for the crystallographic data of CsCdCl$_3$ in this paper is Cambridge Crystallographic Data Center (CCDC): 2313854.

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

## Acknowledgements

This work was supported by the National Natural Science Foundation of China (Grant No. 22275021), the Beijing Municipal Natural Science Foundation (Grant No. L234064), the Beijing Nova Program (Grant No. 20230484414) and the Fundamental Research Funds for the Central Universities.

## Author contributions

Conceptualization: D. Y. and T. C.; Methodology: D. Y. and T. C.; Investigation: T. C.; Supervision: D. Y.; Writing—original draft: T. C.; Writing—review & editing: D. Y.

## Competing interests

The authors declare no competing interest.
