## [Peer Review File NEW · Nature Communications]

Full-color, time-valve controllable and Janus-type long-persistent luminescence from all-inorganic halide perovskitesEditorial Note: Parts of this Peer Review File have been redacted as indicated to remove third-party material where no permission to publish could be obtained.

Reviewer #1 (Remarks to the Author):

In this work, Chen et al. introduced variable concentrations of anions (Br⁻) and cations (Sn²⁺) into hexagonal CdCl₂ all-inorganic perovskites, breaking the original symmetry of the chalcogenides and forming new traps and luminescence centers. The authors emphasize that CsCdCl₃:x%Br exhibits time- and temperature-dependent luminescence, as well as that CsCdCl₃:x%Sn exhibits forward and reverse excitation-dependent luminescence. However, the explanations of the luminescence phenomenon (persistent luminescence) are slightly lacking in the manuscript, and the relatively confusing structure of the manuscript needs to be revised. Therefore, I believe that the manuscript can't be published now due to the following reasons:

1. It is suggested that the authors rearrange the figures in the manuscript according to the order in which they were written, and that they should place the phenomenal figures and the corresponding data in the same section, which will ensure the coherence of the paper.
2. The phenomenon of CsCdCl₃:x%Br having a time-dependent color-changing LPL is only provided in the manuscript with some spectra and decay curves, without any in-depth explanations, which is contrary to the focus of the article mentioned in the title.
3. It is not possible to analyze the difference between the Wet Chemistry and Hydrothermal methods in Figure 1f, so please elaborate or revise the corresponding information.
4. In lines 166-167 of the manuscript, the reference is to the introduction of a cation, as opposed to Br⁻, which is mentioned below; in line 279 of the manuscript, it should be "Supplementary Figs 14-18"; and in lines 281-282 of the manuscript, it should be "Continuous shifting of the excitation to 315 nm". The similar writing problems should be carefully examined and revised by the authors.
5. The correlation between the two materials, CsCdCl₃:x%Br and CsCdCl₃:x%Sn, is insignificant, and is further emphasized by the weak correlation between the two materials by the description used to articulate the two materials in lines 264-265 of the manuscript.
6. The properties of forward and reverse excitation-dependent luminescence of CsCdCl₃:x%Sn are not highlighted in the title of the manuscript.
7. TL analyses are important to illustrate the distribution of traps and are scarce in this work. In Fig. 2, to demonstrate the mechanism of LPL changes color with time, it is recommended to record the TL spectra at each temperature point during the heating process. It is also suggested to provide TL curves monitoring different emission wavelengths to illustrate the dependence of the trap on different luminescence centers. In addition, TL spectra at different pre-irradiation temperatures are needed to illustrate that the traps are continuous. Most of the theoretical calculations in Figure 5 are only available to explain the PL, and the manuscript lacks numerous explanations for the LPL phenomenon.
8. In line 425 of the manuscript, the author mentions that "the final lock code is determined as 1508", provide here the reason for choosing this lock code.
9. Please revise "PLYQ" in Figures 1a and 2i; there are two "d" in the figure caption of Figure 4. Similar formatting issues should be carefully checked and corrected by the authors.

Reviewer #2 (Remarks to the Author):

In this work the authors performed an experimental study of Br⁻ and Sn-doped CsCdCl₃ perovskite materials, with the addition of DFT calculations.

Based on my expertise I can comment mainly the theory part. Overall, the computational part should be described more carefully, and critical aspects/approximations should be discussed/justified, according to the following points. At the current stage it is not possible to recommend publication.

In this respect, the computational setup is standard for materials science applications, and does not neither consider nor mention critical parameters, that affect the predictions. Calculations are done at PBE level, although it is known that the self-interaction error and sometimes spin orbit coupling are important [10.1021/acs.jpcc.1c09594]. This aspect is not mentioned in the main text. Some effort in this respect is suggested.

Also, the computational section is lacking of details, as lattice vectors, cell size and dopant

concentration. It is not specified if the simulated crystal structure corresponds to that determined by single crystal Xrays.

One last concern relies on the take-home message of the simulations. It is unclear the added value of the presented calculations to a very well done and clear experimental work.

Reviewer #3 (Remarks to the Author):

The manuscript entitled "Full-color and time-valve controllable long-persistent luminescence from all-inorganic halide perovskites" authored by Chen et al., demonstrates the development of long persistent luminescence (LPL) materials through the incorporation of Br⁻ and Sn²⁺ ions into hexagonal CsCdCl₃ metal halides. The materials have shown high PLQY and prolonged LPL properties. Additionally, the optical properties of the materials have been applied for high-security anti-counterfeiting and 5D information coding. While the manuscript is interesting, it requires addressing a few issues before being considered for publication.

1. The authors should measure inductively coupled plasma mass spectrometry (ICP-MS) of the Br- and Sn²⁺ doped CsCdCl₃ metal halides to determine the exact dopant percentage. Further, peak deconvolution of the high-resolution XPS spectra in supporting Fig. S4 and S5 are needed.
2. The scales of SEM images in Figure 1 are too small to understand. Please make sure it's comprehensible.
3. In video 1, the materials are polycrystalline. Is there any change in optical properties in the polycrystalline and single crystalline form?
4. The authors should include images of single crystalline CsCdCl₃ (Br- and Sn²⁺ doped) metal halides in the manuscript.
5. "In this work, single crystals of CsCdCl₃ can be grown using a modified hydrothermal reaction." - Authors should cite the literature from where the method is modified.
6. The authors should discuss the effect of electron-phonon coupling with different dopant concentrations and calculate the Huang-Rhys factor.
7. The authors claimed that 5s² electrons of Sn²⁺ have a role in the optical properties of CsCdCl₃:Sn²⁺. In that case, why does CsCdCl₃:Br- exhibit similar thermoluminescence properties (as evidenced in Fig. 2g and Fig. 3g)? Please clarify.

Response to Reviewers

Reviewer #1 (Remarks to the Author):

In this work, Chen et al. introduced variable concentrations of anions (Br^-) and cations (Sn^{2+}) into hexagonal CdCdCl_3 all-inorganic perovskites, breaking the original symmetry of the chalcogenides and forming new traps and luminescence centers. The authors emphasize that $\text{CsCdCl}_3:x\%\text{Br}$ exhibits time- and temperature-dependent luminescence, as well as that $\text{CsCdCl}_3:x\%\text{Sn}$ exhibits forward and reverse excitation-dependent luminescence. However, the explanations of the luminescence phenomenon (persistent luminescence) are slightly lacking in the manuscript, and the relatively confusing structure of the manuscript needs to be revised. Therefore, I believe that the manuscript can't be published now due to the following reasons:

Response: Thank you for reviewer's positive comments. We have tried our best to address the reviewer's concerns, as shown below.

1. It is suggested that the authors rearrange the figures in the manuscript according to the order in which they were written, and that they should place the phenomenal figures and the corresponding data in the same section, which will ensure the coherence of the paper.

Response: Thank you for your valuable comments. In the revised edition, we have reorganized the figures and tables in accordance with the chronological order of article composition. To maintain the beauty and integrity of the figures, few figures are not separated, such as Supplementary Fig. 8 and Supplementary Fig. 15.

2. The phenomenon of $\text{CsCdCl}_3:x\%\text{Br}$ having a time-dependent color-changing LPL is only provided in the manuscript with some spectra and decay curves,

without no in-depth explanations, which is contrary to the focus of the article mentioned in the title.

Response: Thank you for your valuable comments. In this revised version, based on the original steady-state PL spectrum, 2D/3D time-resolved PL spectra, temperature-dependent PL spectral, decay lifetime, PL center and symmetry relation analysis, theoretical calculation, and particularly the latest TL analysis, the phenomenon has been analyzed in multiple ways. And the following conclusions can be further obtained: “Therefore, the controllable color-varying long-afterglow can ultimately be ascribed to the tunable trap states that arise from variations in Br⁻ ion concentration within the broken symmetry skeleton. These new formed traps can influence the decay lifetimes at 482 nm and 595 nm. Upon cessation of excitation, charge carriers stored at the tunable traps can be released and radiated at 482 nm with variable decay times and intensities, subsequently cooperating with the LPL of 595 nm spring from the ideal trap depth, thereby giving rise to this remarkable phenomenon. The associated mechanism is shown in Fig. 5e.”

3. It is not possible to analyze the difference between the Wet Chemistry and Hydrothermal methods in Figure 1f, so please elaborate or revise the corresponding information.

Response: Thank you for your valuable comments. Since temperatures below 200 °C are not sufficient to reach the activation energy required for solid state reactions, Wet Chemistry deviates from the conventional dry synthesis procedure and is more inclined to be classified as a low-temperature wet synthesis approach in "Green chemistry". The Wet Chemistry also covers a variety of methods such as hydrothermal, solvothermal, sonochemical and microwave methods and so on (*Inorg. Chem. Front.*, 2020, 7, 3282–3314). Hydrothermal methods can be extended to solvothermal, glycothermal, alcohothermal, and ammonothermal mechanisms depending on the solvent. As some hydrothermal techniques can be conducted at temperatures exceeding 200 °C, the term "Wet Chemistry" is used in this work to emphasize its environmental friendliness and stark contrast to high-temperature (1000 °C) dry synthesis methods. Secondly, the

capacity to effectively manage operation parameters, like temperature, pressure, purity, stability, and pH, is a core upside of wet chemistry methods for generating well-defined materials with essential aspects (*Int J Hydrogen Energy* 2024, 58, 1190–1203). Hydrothermal methods are more general description. In this work, we find that the relevant products cannot be obtained with a single solvent and the target products can only be obtained with a mixed solvent. And the biggest difference from the conventional hydrothermal approach is the programmed cooling, so that the Wet Chemistry description is closer to the concept that the experimental parameters can be modified.

Page 21, line 508, Methods section, one sentence has been adjusted: The solution was heated at 180 °C for 12 hours in a stainless steel autoclave reactor, and then was programmed cooling to room temperature (RT, referring to 25°C) at a speed of 5 °C/h, which is different from the traditional hydrothermal method by natural cooling.

Fig. 1f has been adjusted as below:

Figure R1. From the latest manuscript of the Fig. 1f.

4. In lines 166-167 of the manuscript, the reference is to the introduction of a cation, as opposed to Br-, which is mentioned below; in line 279 of the manuscript, it should be "Supplementary Figs 14-18"; and in lines 281-282 of the manuscript, it should be "Continuous shifting of the excitation to 315 nm". The similar writing

problems should be carefully examined and revised by the authors.

Response: Thank you for your valuable comments. We have reviewed the full text to avoid such ambiguous writing.

The relevant expression has been adjusted as below,

Page 6, line 165, one sentence has been adjusted: “One inspiration for multi-color LPL is the promise of employing different halide cations with tunable band gaps, as well as incorporating halogen substitution in lead-based halide perovskites, enabling the full-color photoluminescence within nanosecond lifetimes.⁴³”

Page 12, line 316, one sentence has been adjusted: “Continuous shifting of the excitation to 315 nm results in a faint peak at 480 nm, which is negligible in the 2D excitation map (Fig. 4i, Supplementary Fig. 22-26).”

5. The correlation between the two materials, CsCdCl₃:x%Br and CsCdCl₃:x%Sn, is insignificant, and is further emphasized by the weak correlation between the two materials by the description used to articulate the two materials in lines 264-265 of the manuscript.

Response: Thank you for your valuable comments. In fact, the original idea was to study how to achieve LPL. In typical ABX₃ all-inorganic perovskites, the focus primarily revolves around regulating B or X sites. Among them, recent reports have mainly focused on the regulation of LPL around the B site, while achieving LPL from the X site remains largely unexplored. Herein, we want to describe a strategy that the regulation of both the X and B sites can also achieve LPL within a fundamental perovskite structure (CsCdCl₃). During the courses of our experimentation, we encountered a significant challenge: despite numerous attempts to adjust the proportions of various salts, we were unable to get the desired product. However, serendipitously, we discovered that employing a solvent mixing technique yielded favorable outcomes. Subsequently, during the performance testing phase, we encountered these rare phenomena and sought to compile them to present this intriguing work.

Thank you for your good suggestion, the relevant expression has been adjusted as

below:

Page 12, line 297, one sentence has been adjusted: “ Another strategy to anchor the versatility of the LPL involves selecting appropriate metallic cations as activators to provide potential compensating excited states in metal halides.”

6. The properties of forward and reverse excitation-dependent luminescence of CsCdCl₃:x%Sn are not highlighted in the title of the manuscript.

Response: Thank you for your valuable comments. Based on your suggestion, the title of the article has been changed as “Full-color, time-valve controllable and Janus-type long-persistent luminescence from all-inorganic halide perovskites”. The article also has related descriptions: “Specifically, CsCdCl₃:x%Sn exhibit a unique optical behavior analogous to Janus-type emission³³, including both forward and reverse excitation-dependent LPL at low or room temperature, respectively.”

7. TL analyses are important to illustrate the distribution of traps and are scarce in this work. In Fig. 2, to demonstrate the mechanism of LPL changes color with time, it is recommended to record the TL spectra at each temperature point during the heating process. It is also suggested to provide TL curves monitoring different emission wavelengths to illustrate the dependence of the trap on different luminescence centers. In addition, TL spectra at different pre-irradiation temperatures are needed to illustrate that the traps are continuous. Most of the theoretical calculations in Figure 5 are only available to explain the PL, and the manuscript lacks numerous explanations for the LPL phenomenon.

Response: Thank you for your valuable comments. According to your good suggestions, we have further performed and added a more detailed study of the traps in this revised revision.

For the first question, we agree with you that theoretical calculations do primarily describe the PL. One objective of this study is to investigate the PL mechanism through theoretical calculations, aiming to provide a comprehensive explanation for the diverse PL phenomena observed. The relationship between PL and LPL is inherently

intertwined, with LPL being a subsequent formation resulting from trapping and de-trapping process of “PL”. For context, the LPL field of all-inorganic halide perovskites is still in its infancy, and related literatures (such as *Angew. Chem. Int. Ed.* 2022, 61, e202208937; *Angew. Chem. Int. Ed.* 2022, 61, e202210853; *Angew. Chem. Int. Ed.* 2021, 60, 24450–24455; *Adv. Mater.* 2023, 2212022; *Angew. Chem. Int. Ed.* 2023, 62, e202308420) do not involve theoretical calculations to explain PL behaviors. The others also only consider the calculation of traps (such as *Angew. Chem. Int. Ed.* 2022, 61, e202210975) by atoms vacancy. To be honest, it is extremely challenging to consider both doping and vacancy defects simultaneously, and even if it did, it would make the results unreliable and meaningless because of the calculation functional itself. On the other hand, if we consider both doping and vacancy defects at the same time, then we have to calculate $C_{10}^1 \times C_{29}^1 = 290$ structures, which will take 207.1 weeks ($290 \times 5 \div 7 = 207.1$, at least 5 days for one structure during calculation), these are both undesirable and uneconomical. However, this is only for single doping and single vacancy, if we consider multi-doping and multi-vacancy, which will become a cosmic computational process. In this work, we can obtain the relationship between luminescence and the symmetry center of doping through these calculations, which is a valuable and not an easy task.

Therefore, we also agree with your suggestion for further explanation through TL, as follows:

Page 10, line 256, one sentence has been adjusted: “For CsCdCl₃:x%Br, four different cases arise: i) $x = 0.2 \sim 0.5$, the trap energy level (E_{trap}) was mainly determined at 0.74 eV; ii) $x = 0.8 \sim 5$, these E_{trap} were distributed around at 0.73, 0.78 \sim 0.93 eV; iii) $x = 10 \sim 15$, the E_{trap} occurred at 0.71 \sim 0.74 eV, where the E_{trap} can be estimated by Urbach’s empirical formula $E_{\text{trap}} = T_m/500$ (T_m is the temperature of TL peak)⁵¹.”

Page 11, line 266, several sentences have been added: “To further prove this hypothesis and elucidate the phenomenon of color-changing afterglow, “thermal cleaning” method was conducted for CsCdCl₃:0.8%Br and CsCdCl₃:10%Br. As shown in Supplementary Fig. 16a and 17a, the TL intensity decreased while T_m increased with rising excitation temperature, which can be attributed to the redistribution of charge

carriers from deep traps to shallow traps (STs).⁵⁴ The E_{trap} and excitation temperature exhibit a strong linear relationship (Supplementary Fig. 16b and 17b), which directly indicates the presence of a quasi-continuous distribution of trap depths in the shallow region.⁵⁵ The initial-rise method was employed to analyze the shallowest trap levels, wherein the Arrhenius equation was utilized to ascertain that the concentration of trap-carriers remains relatively constant on the low-temperature side of the TL curve (Supplementary Fig. 16c and 17c). It is worth noting that the shallowest trap levels do not converge to the average values of 0.17 eV and 0.41 eV (Supplementary Fig. 16d and 17d), respectively. These results strongly indicate that afterglow activity levels are non-narrowly distributed and beneficial to the formation of the color-changing afterglow.⁵⁶ The shallow trap depths of CsCdCl₃:0.8%Br and CsCdCl₃:10%Br were estimated to be 0.53 eV and 0.73 eV, respectively, based on the Hoogenstraaten method⁵⁷, providing further evidence that the free storage and release of charge carriers from the traps is an important factor in ensuring ultralong-lived emission. Furthermore, the 3D TL spectra are used to analyze color-changing afterglow based on the relationship between the emission and TL spectrum. As shown in Supplementary Fig. 18 c-d and 19 c-d, the emission centers at 595 nm are concentrated at 319 K and 421 K in CsCdCl₃:0.8%Br, and at 327 K and 416 K in CsCdCl₃:10%Br, implying the LPL and anti-thermal quenching properties of 595 nm are empowered by both deep and shallow traps. The emission centers at 482 nm initially manifest at a temperature of 295 K (0.59 eV) in CsCdCl₃:0.8%Br (Supplementary Fig. 18 b), and then can be sustained as the temperature rises to 327 K (0.65 eV) in CsCdCl₃:10%Br (Supplementary Fig. 19 b-c). These results demonstrate that by modulating concentration of Br⁻ ions, the trap at 482 nm can be adjusted closer to the ideal trap depth (0.67 ~ 0.76 eV) for LPL, resulting in a longer afterglow lifetime at 482 nm, which is highly consistent with the luminescent decay curves at 482 nm (Fig. 2h). Therefore, the controllable color-varying long-afterglow can ultimately be ascribed to the tunable trap states that arise from variations in Br⁻ ion concentration within the broken symmetry skeleton. These new forming traps can influence the decay lifetimes at 482 nm and 595 nm. Upon cessation of excitation, charge carriers stored at the tunable traps can be released and radiated at 482 nm with

more variable decay times and intensities, subsequently cooperating with the LPL of 595 nm spring from the ideal trap depth, thereby giving rise to this remarkable phenomenon. The associated mechanism is shown in Fig. 5e.”

Page 15, line 375, several sentences have been added: “Focusing on the TL spectra (Fig. 4g), CsCdCl₃:x%Sn exhibit distinct trap energy levels (estimated by $E_{\text{trap}} = T_m/500$) with different x values, where they all display shallow trap energy levels around 0.68~0.71 eV, as well as deep trap energy levels ranging from 0.79 to 0.90 eV with $x = 3 \sim 15$. It is worth noting that the TL properties of CsCdCl₃:x%Sn are similarly to those of CsCdCl₃:x%Br, attributed to the formation of trap states caused by similar Cl, Cs, and Cd ions vacancies or lattice dislocations in the broken CsCdCl₃ skeleton during Sn²⁺ or Br⁻ ion doping. Given this, it is no surprise that traps play a crucial role in saving charge carriers from the excited state, evoking the LPL of these CsCdCl₃:x%Sn by releasing charge carriers from shallow traps or thermally breaking the energy barrier for migration from deep traps, and then to the ground state. To further verify this assumption, we also employ the "thermal cleaning" method to investigate the TL of CsCdCl₃:10%Sn. As illustrated in Supplementary Fig. 41 a-b, the excitation temperature increment leads to the redistribution of charge carriers by facilitating their transfer from deep to shallow traps. Furthermore, the well-established correlations between trap energy and excitation temperature elucidate the quasi-continuous distribution of trap depths. These shallowest trap levels do not converge to an average value of 0.14 eV (Supplementary Fig. 41 c-d), indicating the presence of multiple afterglow activity levels that provides the possibility of multiple LPL in this system, herein supporting the LPL of 565 and 595 nm. The shallow trap depths of CsCdCl₃:10%Sn can be determined to be 0.51 eV (Supplementary Fig. 41 e). The 3D TL spectra exhibit three TL centers at 353 K, 390 K and 423 K (Supplementary Fig. 42), respectively. Obviously, the wavelength-resolved TL peak at 353 K (0.71 eV) is broader than that at 390 K (0.78 eV) and 423 K (0.85 eV), and it can be well fitted as two peaks by a Gaussian function. This indicates that both emissions at 565 and 595 nm can be caused by an ideal shallow trap, leading to the formation of LPL. Hence, these results are in good agreement with the experimental phenomena and the proposed

mechanism (Fig. 5f).”

Supplementary Fig. 16 a Results of the partial cleaning ($T_{\text{max}} - T_{\text{exc}}$) experiments. The TL glow curves of CsCdCl₃:0.8%Br were recorded after thermal cleaning at various excitation temperature (T_{exc}). b Dependence of E_{trap} on excitation temperature. c The dependence of $\ln(I(T))$ on $1/kT$, the blue fitting line was determined using the initial rise method. d The shallowest trap distribution of the CsCdCl₃:0.8%Br. e Estimation of

trap depth with the Hoogenstraaten method. Hoogenstraaten's peak position characterization is best known and can be expressed as follows:

$$\frac{\beta_h \cdot \varepsilon}{k_B T_m^2} = s \cdot \exp\left(\frac{-\varepsilon}{k_B T_m^2}\right) \quad \text{S2}$$

where β_h is the heating rate (in K s^{-1}), ε is the trap depth (in eV), k_B is the Boltzmann constant, T_m is the peak temperature in the TL glow curves, and s is the frequency factor (in s^{-1}).

Supplementary Fig. 17 a Results of the partial cleaning ($T_{\text{max}} - T_{\text{exc}}$) experiments. The TL glow curves of $\text{CsCdCl}_3:10\%\text{Br}$ were recorded after thermal cleaning at various excitation temperature (T_{exc}). **b** Dependence of E_{trap} on excitation temperature. **c** The dependence of $\ln(I(T))$ on $1/kT$, the blue fitting line was determined using the initial

rise method. **d** The shallowest trap distribution of the CsCdCl₃:10%Br. **e** Estimation of trap depth with the Hoogenstraaten method.

Supplementary Fig. 18 a 3D TL spectra of CsCdCl₃:0.8%Br as a function of emission wavelength and temperature after UV (254 nm) illumination for 1 min. Wavelength-resolved TL spectra of CsCdCl₃:0.8%Br were monitored at **b** 295K, **c** 319 K and **d** 421 K.

Supplementary Fig. 19 a 3D TL spectra of CsCdCl₃:10%Br as a function of emission wavelength and temperature after UV (254 nm) illumination for 1 min. Wavelength-

resolved TL spectra of CsCdCl₃:10%Br were monitored at **b** 295K, **c** 327 K and **d** 416 K.

Supplementary Fig. 41 **a** Results of the partial cleaning ($T_{\text{max}} - T_{\text{exc}}$) experiments. The TL glow curves of CsCdCl₃:10%Sn were recorded after thermal cleaning at various excitation temperature (T_{exc}). **b** Dependence of E_{trap} on excitation temperature. **c** The dependence of $\ln(I(T))$ on $1/kT$, the blue fitting line was determined using the initial

rise method. **d** The shallowest trap distribution of the CsCdCl₃:10%Sn. **e** Estimation of trap depth with the Hoogenstraaten method.

Supplementary Fig. 42 a 3D TL spectra of CsCdCl₃:10%Sn as a function of emission wavelength and temperature after UV (254 nm) illumination for 1 min. Wavelength-resolved TL spectra of CsCdCl₃:10%Sn were monitored at **b** 353 K, **c** 390 K and **d** 423 K.

8. In line 425 of the manuscript, the author mentions that "the final lock code is determined as 1508", provide here the reason for choosing this lock code.

Response: Thank you for your suggestions. Each lock code represents the result of the selection algorithm. 1508 is obtained by the algorithm of "Time code (at 18s) - QR code (at 18s) - color code (at 18s)". This is also the most important advantage of discoloration afterglow materials, that we can set our own algorithm as security code in the relevant program, and the monochrome afterglow materials can only be used as code. The relevant scenario diagrams are given below. For example, if we want to change time code at 10s and adjust the code sequence, where we can set the algorithm as "QR code (at 18s)-Time code (at 10s) - color code (at 18s)", and the security code will be "843",

other people can only get enormous security codes due to the lack of the algorithm.

The relevant expression has been adjusted as below,

Page 19, line 466, one sentence has been adjusted: “According to algorithm that we have set with ‘Time code (at 18s) - QR code (at 18s) - color code (at 18s)’, the final lock code is determined as “1508” (Supplementary Fig. 57a).”

Figure R2. From the latest manuscript of the Fig. 6f.

Supplementary Figs 57. a-c Conceptual diagram of 3D code reader and simple algorithm setup process. Note: Leveraging 5G communications, one can establish algorithms across continents to ensure access rights to top-secret documents under their disposal.

9. Please revise "PLYQ" in Figures 1a and 2i; there are two "d" in the figure caption of Figure 4. Similar formatting issues should be carefully checked and

corrected by the authors.

Response: Thank you for your comments. These typos have been corrected in the new revised version:

Figure R3. From the latest manuscript of the Fig. 1.

Figure R4. From the latest manuscript of the Fig. 2.

Figure R5. From the latest manuscript of the Fig. 4.

Reviewer #2 (Remarks to the Author):

In this work the authors performed an experimental study of Br- and Sn-doped CsCdCl₃ perovskite materials, with the addition of DFT calculations. Based on my expertise I can comment mainly the theory part. Overall, the computational part should be described more carefully, and critical aspects/approximations should be discussed/justified, according to the following points. At the current stage it is not possible to recommend publication.

Response: We are very appreciated to the Reviewer's evaluation and comments. We have tried our best to address the reviewer's concerns, as shown below.

In this respect, the computational setup is standard for materials science

applications, and does not neither consider nor mention critical parameters, that affect the predictions. Calculations are done at PBE level, although it is known that the self-interaction error and sometimes spin orbit coupling are important [10.1021/acs.jpcc.1c09594]. This aspect is not mentioned in the main text. Some effort in this respect is suggested.

Response: Thank you for your valuable comments. As noted by reviewer, it is well known that PBE functionals may underestimate the band gap [*J. Am. Chem. Soc.* 2009, 131, 2, 816–825], and we also expect improvement of accuracy by using HSE06 or HSE06/SOC functionals as recommended in the literature [10.1021/acs.jpcc.1c09594]. However, the HSE06 or HSE06/SOC are time-consuming and costly to run even on large commercial computer clusters. Currently, our laboratory is equipped with limited computing resources (CPU: Intel E5-2696V3), which somewhat hinders the realization of such ambitious goals. In fact, a substantial body of relevant literatures have emerged (such as *J. Am. Chem. Soc.* 2024, 146, 1167–1173; *J. Am. Chem. Soc.* 2023, 145, 14112–14123; *Joule*, 2022, 7, 2595 – 2608; *Chem* 2022 ,8, 3051-3063; *Sci Adv.*, 2020,6, eaaw7453, etc.) based on the implementation of PBE level. Furthermore, based on your good suggestion, we compared the two methods by using PBE and PBE+SOC. As shown in the Figure R6 below, the band gaps and accuracy are further underestimated by using PBE+SOC in both examples of $D_{3d}-Br_1-C_{3v}$ and the $C_{3v}-Sn_2$ models, which are also consistent with the conclusion from this previous literature [10.1021/acs.jpcc.1c09594] that “Based on an error analysis, we found that the addition of SOC to band gaps computed at the PBE level overcorrects the gaps and induces an overall deterioration of the accuracy”. Second, selecting the PBE/SOC levels in the same model takes about 2.4 days longer than using only the PBE levels in Quantum ESPRESSO (qe-7.2). Therefore, we sincerely appreciate your good suggestion again. While spin-orbit coupling is sometimes important in more heavy atomic systems such as Pb-containing perovskites, so we think that the original PBE level may be more suitable for current systems.

Figure R6. Comparison of calculation methods using (a,c) PBE and (b,d) PBE/SOC functionals in $D_{3d}\text{-Br}_1\text{-C}_{3v}$ and $C_{3v}\text{-Sn}_2$, respectively.

Also, the computational section is lacking of details, as lattice vectors, cell size and dopant concentration. It is not specified if the simulated crystal structure corresponds to that determined by single crystal Xrays. One last concern relies on the take-home message of the simulations. It is unclear the added value of the presented calculations to a very well done and clear experimental work.

Response: Thank you for your helpful comments. According to your valuable suggestions, we added the descriptions of lattice vectors, cell size and dopant concentration in the completed file, and added Supplementary Table 5 in the revised Supporting Information. In this study, we aimed to investigate the photoluminescence (PL) behaviors of CsCdCl_3 perovskites *via* doping with Br^- or Sn^{2+} ions in different symmetric centers, thereby disrupting local symmetry and inducing distinct PL properties. Regarding the doping concentration, we believe our doping models can accommodate different symmetries in CsCdCl_3 perovskites. And constructing larger

supercells to achieve a much smaller doping models would be excessively time-consuming and may not necessarily yield a substantial improvement in accuracy and results, as this aspect is more reliant on functional level. Importantly, our computational results are strongly supported by experimental findings. For context, the field of all-inorganic halide perovskites with long-persistent afterglow is still in its infancy, and related literatures (such as *Angew. Chem. Int. Ed.* 2022, 61, e202208937; *Angew. Chem. Int. Ed.* 2022, 61, e202210853; *Angew. Chem. Int. Ed.* 2021, 60, 24450–24455; *Adv. Mater.* 2023, 2212022; *Angew. Chem. Int. Ed.* 2023, 62, e202308420) have not involved theoretical calculation to explain PL behaviors. To be frank, we have exerted utmost effort and dedicated considerable time to the calculations. However, achieving absolute precision through the utilization of more expensive functional level would indeed present challenges.

The following additions have been added in the main text:

Page 17, line 405, one sentence has been adjusted: “To gain deeper insights into the electronic structure and luminescence mechanisms by disrupting the original symmetry in the all-inorganic skeleton, ten models of CsCdCl₃ doped with Br and Sn were constructed (Supplementary Fig. 43).”

Page 17, line 407, one sentence has been added: “Their lattice constants (V) exhibit an enlargement to further confirm the expansion of the host lattice in XRD patterns (Supplementary Table 5, Supplementary Fig. 4-5).”

Page 22, line 539, theoretical calculations section, three sentence for doing models have been adjusted: “The computational models encompass all the potential sites by disrupting original symmetry in all-inorganic skeletons through doping with Br⁻ or Sn²⁺ ions. Such as CsCdCl₃:5.5%Br model was constructed by replacing one Cl atom with a Br atom in a 30-atom cell model, which is analogous to the CsCdCl₃:5%Br model. Similarly, the CsCdCl₃:16.6%Sn model, achieved by substituting one Cd atom with a Sn atom, which is analogous to the CsCdCl₃:15%Sn.”

Page 22, line 545, theoretical calculations section, one sentence has been added: “The Γ point was used to represent the Brillouin zone.”

Page 22, line 547, theoretical calculations section, one sentence has been added:

“The lattice constants were optimized and compared with experimental results in Supplementary Table 5.”

Supplementary Table 5 Lattice constants of Br⁻ or Sn²⁺ -doped CdCdCl₃ perovskites

Models	a(Å)	b(Å)	c(Å)	V(Å ³)	α°	β°	γ°
D _{3d} -Br ₁ -C _{3v}	7.5694	7.5996	18.8776	941.6613	89.5989	90	119.8684
D _{3d} -Br ₂ -C _{3v}	7.5996	7.5996	18.8776	941.6632	90.4010	89.5990	120.2628
D _{3d} -Br ₃ -C _{3v}	7.5996	7.5694	18.8771	941.6637	90	90.4018	119.8687
C _{3v} -Br ₄ -C _{3v}	7.5758	7.5749	18.8678	937.7418	90	90	119.9958
C _{3v} -Br ₅ -C _{3v}	7.5748	7.5757	18.8675	937.6839	90	90	119.9958
C _{3v} -Br ₆ -C _{3v}	7.5758	7.5758	18.8671	937.6753	90	90	120.0093
D _{3d} -Sn ₁	7.6218	7.6218	19.0540	958.6043	90	90	120
C _{3v} -Sn ₂	7.6044	7.6044	19.0800	955.5170	90	90	120
C _{3v} -Sn ₃	7.6040	7.6040	19.0821	955.5352	90	90	120
D _{3d} -Sn ₄	7.6240	7.6240	19.0473	958.8152	90	90	120
Experimental	7.3797	7.3797	18.3778	866.76	90	90	120

Reviewer #3 (Remarks to the Author):

The manuscript entitled "Full-color and time-valve controllable long-persistent luminescence from all-inorganic halide perovskites" authored by Chen et al., demonstrates the development of long persistent luminescence (LPL) materials through the incorporation of Br⁻ and Sn²⁺ ions into hexagonal CsCdCl₃ metal halides. The materials have shown high PLQY and prolonged LPL properties. Additionally, the optical properties of the materials have been applied for high-security anti-counterfeiting and 5D information coding. While the manuscript is interesting, it requires addressing a few issues before being considered for publication.

Response: Thank you for reviewer’s positive comments. We have tried our best to address the reviewer’s concerns, as shown below.

1. The authors should measure inductively coupled plasma mass spectrometry (ICP-MS) of the Br⁻ and Sn²⁺ doped CsCdCl₃ metal halides to determine the exact

dopant percentage. Further, peak deconvolution of the high-resolution XPS spectra in supporting Fig. S4 and S5 are needed.

Response: Thank you for your valuable comments. According to your suggestions, we have consulted the analysis and test center of our university, and the instructor of experimental operator shows: “although their ICP could measure more than 60 elements, it could not measure H, C, N, O, P, halogens, inert gas elements and actinides. ICP-MS has a very low detection limit; the detection limit of its solution is mostly at the ppt level. However, due to the poor salt tolerance of ICP-MS, the actual detection limit can be up to 50 times worse. Some light elements also cause serious interference in ICP-MS, resulting in a poor actual detection limit.” Even though we cannot measure Br, we still want to measure Sn²⁺ doped CsCdCl₃. After planning to prepare its solution, we discovered that even after subjecting Sn²⁺ doped CsCdCl₃ to ultrasonic treatment at 60 °C for two hours in ultra-pure water containing 50% concentrated hydrochloric acid, it remained insoluble while still exhibiting excellent luminous properties. Furthermore, we attempted to dissolve it in concentrated nitric acid (Figure R7). However, our efforts were unsuccessful. Finally, the ICP test plan also failed because it could not form its clarified solution for testing. At present, we have no idea to complete your suggestion. In fact, the powder X-ray diffraction data combined with EDS for double quantitative element analysis that we employed is also primarily based on the literature sources (such as *Angew. Chem. Int. Ed.* 2022, 61, e202210853; *Nature*, 2018, 563, 541–545; *J. Am. Chem. Soc.* 2020,142,10780–10793; *J. Phys. Chem. C* 2021, 125, 1954–1962; *Sci.Rep.* 2016,6, 19746). It can be seen from the results that the calculated value of XPS on concentration is roughly consistent with that of EDS (Supplementary Table 2-3). This double quantitative elemental analysis is also effective compared to that only using XPS or EDS value. We greatly appreciate your suggestion. During the experiment, we discovered that our material possesses a certain level of waterproofness and resistance to hydrochloric acid corrosion, which is undeniably an exciting characteristic. Therefore, we have added the following description in the revised version.

Page 14, line 362, one sentence has been added: “Interestingly, this kind of materials

can maintain high luminous intensity in ultra-pure water and acid (Supplementary Fig. 36).”

Furthermore, according to your suggestions, peak deconvolution of the high-resolution XPS spectra in supporting Fig. S4 and S5 has been adjusted, as shown below.

Figure R7. Some experimental information related to ICP

Supplementary Fig. 36. a-g The optical phenomenon of $\text{CsCdCl}_3:5\%\text{Br}$ and $\text{CsCdCl}_3:10\%\text{Sn}$ were observed in ultra-pure H_2O and HCl (v:v=1:1) solution by ultrasonic treatment at 60°C for two hours, respectively. Note: Supplementary Fig. 36 a, the centrifuge tubes from left to right are $\text{CsCdCl}_3:5\%\text{Br}$, $\text{CsCdCl}_3:5\%\text{Br}$, $\text{CsCdCl}_3:10\%\text{Sn}$ and $\text{CsCdCl}_3:10\%\text{Sn}$, respectively.

Figure R8. From the latest Supplementary Information of the Supplementary Fig. 2.

Figure R9. From the latest Supplementary Information of the Supplementary Fig. 3.

2. The scales of SEM images in Figure 1 are too small to understand. Please make sure it's comprehensible.

Response: Thank you for your valuable comments. According to your good suggestions, we have adjusted the following Fig. 1 in the revised version.

Figure R10. From the latest manuscript of the Fig. 1.

3. In video 1, the materials are polycrystalline. Is there any change in optical properties in the polycrystalline and single crystalline form?

Response: Thank you for your valuable comments. We have checked the video many times, and we understand that you mean that the polycrystalline state is also formed by single crystals lying together. Some crystals may be broken by the scraping spoon when they are removed from the reactor. Therefore, their optical properties are identical. This point can also be reflected from the spectral data, if their optical properties are not the same, then there will be a lot of sub-peaks on the spectrum, rather than showing consistency. “Upon alternation by Br^- or Sn^{2+} ions, the powdered X-ray diffraction (PXRD) patterns of $\text{CsCdCl}_3:x\%\text{Br}$ and $\text{CsCdCl}_3:x\%\text{Sn}$ closely agree with the pattern in the PDF#18-0337, confirming the single-phase purity of the synthesized Br^- or Sn^{2+} -doped CsCdCl_3 (Fig. 1b, c and Supplementary Fig. 4-5).” In addition, the video only briefly showcases their optical performance and the difference we perceive with our naked eye (576 million pixels) is significant due to limitations of the camera itself and

memory constraints imposed by the journal. The original video was compressed from 3.51GB to 59.1MB, resulting in severe frame loss and thus unsatisfactory rendering. The purpose of our video presentation is just to outline the luminescence phenomenon.

Figure R11. The original video was compressed from 3.51GB to 59.1MB.

4. The authors should include images of single crystalline CsCdCl₃ (Br⁻ and Sn²⁺ doped) metal halides in the manuscript.

Response: Thank you for your valuable comments. According to your good suggestions, we added images of single crystalline CsCdCl₃, CsCdCl₃:x%Br and CsCdCl₃:x%Sn into supporting information. As we can see, not all crystal forms are complete because the spoon may break a crystal when it is removed from a high-pressure reactor.

Page 6, line 145, one sentence has been adjusted: “Scanning electron microscope (SEM) images show a typical spindle shape of Br⁻ or Sn²⁺-doped CsCdCl₃ crystals (Supplementary Figs. 27),”

Supplementary Fig. 6. SEM images of single crystalline **a** CsCdCl_3 , **b** $\text{CsCdCl}_3:x\%\text{Br}$ and **c** $\text{CsCdCl}_3:x\%\text{Sn}$.

5. "In this work, single crystals of CsCdCl_3 can be grown using a modified hydrothermal reaction." - Authors should cite the literature from where the method is modified.

Response: Thank you for your valuable comments. we have added related literature (*ACS Appl. Mater. Interfaces* 2023, 15, 24629–24637; *Angew. Chem.Int. Ed.* 2022, 61, e20220893) in this revised version.

Page 21 line 508, Methods section, one sentence has been adjusted: “The solution was heated at 180 °C for 12 hours in a stainless steel autoclave reactor, and then was programmed cooling to room temperature (RT, referring to 25°C) at a speed of 5 °C/h, which is different from the traditional hydrothermal method by natural cooling.”

6. The authors should discuss the effect of electron-phonon coupling with different dopant concentrations and calculate the Huang-Rhys factor.

Response: Thank you for your valuable comments. According to your good suggestions, we have discussed the impact of electron-phonon coupling and included the Huang-Rhys factor figures in the revised versions.

The following discuss have been adjusted in the main text:

Page 8, line 189, one sentence has been adjusted: “To comprehend this behavior, we analyze the structure-luminescence relationship, the newly formed $[\text{CdCl}_{6-n}\text{Br}_n]^{4-}$ would become a distorted octahedron due to the distinct bond lengths of Cd–Cl (2.66 Å) and Cd–Br (2.71 Å)⁴⁷, which may promote STEs at room temperature by the lattice distortion in excited states.⁴⁸⁻⁵⁰”

Page 9, line 210, several sentences have been adjusted: “In Supplementary Fig. 11, the Huang-Rhys factor (S) for 482 nm and 595 nm have been enhanced from 45 to 66 and 12 to 19 by increasing the doping concentrations of Br^- ions from 0.8% to 10%, respectively. The changed value of S indicates that increasing the doping concentrations of Br^- ions can lead to greater lattice distortion in D_{3d} and C_{3v} symmetry and enhance photon-phonon coupling, then forming different optical properties from pure CsCdCl_3 perovskites. Furthermore, such large S and significant Stokes-shift, provide direct evidence that the LPL of band 2 and band 1 are associated with STEs^{47,50}.”

Page 14, line 339, several sentences have been adjusted: “The Huang-Rhys factor (S) at 595 nm has been determined to be ranged from 16 to 25 (Supplementary Fig. 30a), owing to the increased doping concentrations of Sn^{2+} ions which augment the $[\text{SnCdCl}_9]^{5-}$ moieties in C_{3v} symmetry. These results directly demonstrate that band i is also associated with STEs due to the elevated proportion of Cd^{2+} -based octahedral distortion.”

Page 14, line 351, one sentence has been adjusted: “(b) such high Huang-Rhys factor (S) (Supplementary Fig. 30b) and large Stokes shift crucially support its STEs⁶³.”

Supplementary Figs. 11. FWHM of **a** $\text{CsCdCl}_3:0.8\%\text{Br}$ and **b** $\text{CsCdCl}_3:10\%\text{Br}$ at 482 nm and 595 nm versus temperature from the delayed spectra of Supplementary Fig. 10,

respectively.

The electron-phonon coupling effect of Br-doped CsCdCl₃ was discussed by fitting FWHM versus temperature according to follow Equation S1:

$$\text{FWHM} = 2.36\sqrt{S}\hbar\omega_{\text{phonon}}\sqrt{\coth\frac{\hbar\omega_{\text{phonon}}}{2k_{\text{B}}T}} \quad \text{S1}$$

Here, k_{B} is the Boltzmann constant, and S is the Huang-Rhys electron-phonon coupling parameter, and $\hbar\omega_{\text{phonon}}$ is the effective phonon energy.

Supplementary Fig. 30. FWHM of **a** CsCdCl₃:3%Sn and **b** CsCdCl₃:10%Sn at 595 nm and 565 nm versus temperature from the delayed spectra of Supplementary Fig. 27-28, respectively.

The electron-phonon coupling effect of Sn-doped CsCdCl₃ was discussed by fitting FWHM versus temperature according to Equation S1.

7. The authors claimed that 5s² electrons of Sn²⁺ have a role in the optical properties of CsCdCl₃:Sn²⁺. In that case, why does CsCdCl₃:Br⁻ exhibit similar thermoluminescence properties (as evidenced in Fig. 2g and Fig. 3g)? Please clarify.

Response: Thank you for your valuable comments. We understand your concern. In fact, in the same CsCdCl₃ skeleton, due to doping with Sn²⁺ or Br⁻ ions, its original structure is disrupted, resulting in the presence of atomic Cl, Cs and Cd vacancies in the same symmetry and leading to the formation of similar trap levels. Correspondingly, these similar trap levels behave similarly on the thermoluminescence spectrum. The charge carriers from the 5s² of Sn²⁺ will also be replenished and stored by the corresponding

trap levels. Upon cessation of the UV lamp, these charge carriers are released from trap levels and form their relevant long persistent luminescence.

Page 15, line 378, one sentence has been added: “It is worth noting that the TL properties of CsCdCl₃:x%Sn are similarly to those of CsCdCl₃:x%Br, attributed to the formation of trap states caused by similar Cl, Cs, and Cd ions vacancies or lattice dislocations in the broken CsCdCl₃ skeleton during Sn²⁺ or Br⁻ ion doping.”

Thank you again for all of the very useful suggestions above.

Reviewer #1 (Remarks to the Author):

In this work, Chen et al. introduced variable concentrations of anions (Br⁻) and cations (Sn²⁺) into hexagonal CdCl₂ all-inorganic perovskites, breaking the original symmetry of the chalcogenides and forming new traps and luminescence centers. The authors emphasize that CsCdCl₃:x%Br exhibits time- and temperature-dependent luminescence, as well as that CsCdCl₃:x%Sn exhibits forward and reverse excitation-dependent luminescence. After the authors have revised the manuscript, it is slightly lacking in the addition of the LPL phenomenon and the logic of the article. Therefore, I suggest that the manuscript can be published in 《Nature Communications》 after minor revisions. The specific feedback is given below:

1. The authors should include the time-dependent LPL spectra of the CsCdCl₃:x%Br and the corresponding CIE in the manuscript.
2. The author's explanation for the lack of coherence between the two materials still remains inadequate, and the sentence in Page 12, line 297 is more analogous to the articulation of the two articles.

Reviewer #2 (Remarks to the Author):

The authors have made relevant improvements in the revised version of the manuscript.

- I must say that the theory part is now self standing, but it is still hard to extract its added value. However, the simulated details are however clearer now.

- The doping ratio is in some cases very large (up to 16%), which is closer to a compositional change rather than doping.

Reviewer #3 (Remarks to the Author):

In the revised manuscript, the authors have responded to a few queries. However, there are still a few concerns that need to be addressed:

a) In the case of CsCdCl₃:10Sn²⁺, a notable disparity in XPS peak intensity persists between Sn²⁺ and Cd²⁺. Why does the intensity of Sn²⁺ appear significantly lower compared to Cd²⁺?

b) In the context of ICP-MS sample preparation, authors can dissolve the CsCdCl₃:10Sn²⁺ powder (ground the crystals to powder) in aqua regia and dilute it in water.

c) It is suggested that the authors include images depicting the as-synthesized single crystals rather than SEM images. Notably, the SEM image in Fig. S6 (X=10) appears similar to polycrystals.

d) The calculation of the Huang-Rhys factor indicates an increase in electron-phonon coupling with heightened Sn²⁺ dopant concentration. However, it remains ambiguous whether increased defects give rise to more STEs, thereby leading to persistent luminescence, or the charge carriers from the 5s² orbital of Sn²⁺ contribute to the trap states to generate persistent luminescence.

Point-by-Point Response to Review Comments

We sincerely appreciate the valuable suggestions from editors and reviewers. We have endeavored to improve the quality of the manuscript following reviewers' comments. All modifications have been highlighted with yellow color in the revised manuscript.

Reviewer #1 (Remarks to the Author):

In this work, Chen et al. introduced variable concentrations of anions (Br^-) and cations (Sn^{2+}) into hexagonal CdCdCl_3 all-inorganic perovskites, breaking the original symmetry of the chalcogenides and forming new traps and luminescence centers. The authors emphasize that $\text{CsCdCl}_3:x\%\text{Br}$ exhibits time- and temperature-dependent luminescence, as well as that $\text{CsCdCl}_3:x\%\text{Sn}$ exhibits forward and reverse excitation-dependent luminescence. After the authors have revised the manuscript, it is slightly lacking in the addition of the LPL phenomenon and the logic of the article. Therefore, I suggest that the manuscript can be published in 《Nature Communications》 after minor revisions. The specific feedback is given below:

Response: We express our gratitude for the reviewer's dedicated time and effort in providing feedback on our manuscript. We have revised the manuscript according to your valuable comments and suggestions.

1. The authors should include the time-dependent LPL spectra of the $\text{CsCdCl}_3:x\%\text{Br}$ and the corresponding CIE in the manuscript.

Response: We thank the reviewer for the valuable comment. According to your helpful suggestions, the time-dependent emission spectra and the corresponding CIE of $\text{CsCdCl}_3:0.2\%\text{Br}$, $\text{CsCdCl}_3:0.5\%\text{Br}$ and $\text{CsCdCl}_3:0.8\%\text{Br}$ have been further added into the revised manuscript.

The relevant expression has been adjusted as below,

Page 9, line 242, one sentence has been added: “Time-dependent emission spectra and the corresponding CIE of $\text{CsCdCl}_3:0.2\sim 0.8\%\text{Br}$ also show that the color can be

changed by decay time (Supplementary Fig. 15).”

Supplementary Fig. 13 Time-dependent emission spectra and the corresponding CIE of **a-b** CsCdCl₃:0.2%Br, **c-d** CsCdCl₃:0.5%Br and **e-f** CsCdCl₃:0.8%Br.

2. The author's explanation for the lack of coherence between the two materials still remains inadequate, and the sentence in Page 12, line 297 is more analogous to the articulation of the two articles.

Response: We thank the reviewer for the valuable comment. We are moved by the excelsior spirit from the reviewer, and tried our best to address this explanation.

To highlight the connection of the two materials, the relevant expression has been adjusted as below:

Page 12, line 304, one sentence has been adjusted: “The same host material exhibits a variety of luminous capabilities, offering potential for multifunctional applications. In the case of controlled color-changing afterglow achieved by Br⁻ ions doping engineering, we continue to extend luminescence functionality of the CsCdCl₃ host and choose the 5s² electronic configuration of Sn²⁺ ions as activators.”

Reviewer #2 (Remarks to the Author):

The authors have made relevant improvements in the revised version of the manuscript.

Response: We appreciate the reviewer’s insightful and helpful comments on our manuscript.

- I must say that the theory part is now self standing, but it is still hard to extract its added value. However, the simulated details are however clearer now.

Response: We thank the reviewer for the thoughtful and valuable comment. DFT calculation has indeed supplied new understanding on the solid-state optical properties from an electronic structure view, which may be not obtained directly from experimental characterization. For example, the computational result shows that doping of Cl⁻ and Sn²⁺ can affect the change of skeleton luminescence center. DOS and PDOS also show the contribution of Br 4p and Sn 5s²/5p. The charge density maps intuitively show the unique contribution of Br⁻ and Sn²⁺ doping in D_{3d} and C_{3v} symmetry, which leads to the difference in luminescence. Importantly, the emission of 482 nm from the [CdCl_{6-n}Br_n]⁴⁻ moiety in both D_{3d} and C_{3v} symmetry have been mutually verified by both theoretical calculations and experiments. Cs orbitals play a negligible role in the band structures of these models, demonstrating that typical ABX₃ all-inorganic perovskites with multiple luminous capabilities can be realized by focusing the B or X sites. In addition, the reverse excitation-dependent behavior observed in experiment, can also be proved by theoretical calculation that the 565 nm is attribute to the luminescence center of Sn²⁺ ions in C_{3v} [SnCdCl₉]⁵⁻, this result is very important to support our study. Therefore, the combination of theory part could supply new insight and better understanding on how the ions doping could influence of the luminescence

color of the CsCdCl₃ from the viewpoint of electronic structure.

To further highlight the important of combined experimental and computational studies, the senescence in the abstract has been further adjusted:

Page 2, line 32, one sentence has been rephrased: Combination of both experimental and computational studies, this work not only introduces a local-symmetry breaking strategy for simultaneously enhancing afterglow lifetime and efficiency, but also provides new insights into the multimode LPL materials for applications in luminescence, photonics, and information storage.

Page 16, line 407, one sentence has been rephrased: To gain deeper insights into the electronic structure and color-tunable luminescence mechanisms by disrupting the original symmetry and the doping effect in the all-inorganic skeleton, DFT calculations were performed on ten idealized models of CsCdCl₃ doped with Br and Sn (Supplementary Fig. 45).

- The doping ratio is in some cases very large (up to 16%), which is closer to a compositional change rather than doping.

Response: We thank the reviewer for the valuable comment. All along, the purpose of our theoretical calculation has not mainly focused on the effect of doping concentration on luminescence. The purpose of establishing these calculation models is to investigate the impact of Cl⁻ and Sn²⁺ on luminescence at different doping sites and symmetry centers. In the Figure R1, Sn atoms are doped in different centers of symmetry, also refer to relevant literature [*Angew. Chem. Int. Ed.* 2022, 61, e202210975; *Adv. Optical Mater.* 2023, 2300323.] on doping modeling. In addition, we also see that the establishment of models requires a certain unity to facilitate comparison, and the results of different models are not comparable, as well as the intentional regulation of doping ratios by using different models. We also benefit from using these unified models, such as the finding of the 482 nm from the [CdCl_{6-n}Br_n]⁴⁻ moiety in both D_{3d} and C_{3v} symmetry, and the 565 nm from Sn²⁺ ions in C_{3v} [SnCdCl₉]⁵⁻. If we don't build these unified models, the aforementioned findings would be difficult to obtain. We have also tried to pursue the reduction of the element doping concentration by building up the

most basic supercells ($2 \times 2 \times 2$: 180 atoms, 2.7% Sn; or even $2 \times 1 \times 1$: 60 atoms, 8.3% Sn) or reducing the element proportion. In those cases, there are some difficult calculation errors, which may be related to the computing software and the computing server itself (Figure R2). We would like to emphasize that the doping purity is all single-phase purity, which can be proved here by XRD results. Finally, we also hope that the reviewer can see the efforts we have made.

Figure R1. Sn atom is doped in different centers of symmetry.

```

1 | kernel:NMI watchdog: BUG: soft lockup - CPU#2 stuck for 23s! [polkitd:966]
2 | kernel:NMI watchdog: BUG: soft lockup - CPU#2 stuck for 22s! [swapper/2:0]
3 | .....
   | Error in routine c_bands (1):
   | too many bands are not converged
   | .....
   |
   | stopping ...
   | .....

```

Figure R2. The calculation process encountered an error.

Reviewer #3 (Remarks to the Author):

In the revised manuscript, the authors have responded to a few queries. However, there are still a few concerns that need to be addressed:

Response: We express our gratitude for the reviewer's dedicated time and effort in providing feedback on our manuscript. We have revised the manuscript according to your valuable comments and suggestions.

a) In the case of CsCdCl₃:10Sn²⁺, a notable disparity in XPS peak intensity persists between Sn²⁺ and Cd²⁺. Why does the intensity of Sn²⁺ appear significantly lower compared to Cd²⁺?

Response: We thank the reviewer for this valuable comment. X-ray photoelectron spectroscopy (XPS) is more likely a surface analysis technology of materials (its detection depth is generally less than 10 nm), which can not only characterize the types and semi-quantitative content of elements on the surface of materials, but also give their chemical valence states, chemical bonds and related electronic structure information. The key of XPS quantitative analysis is to convert the detected signal intensity (based on Einstein's photoemission equation) into element content. However, due to the complex factors (such as detection depth, physical adsorption, chemical adsorption, and the inability to maintain sample surface stability), its peak intensity is not simply and proportionally related to the element content. We also know that some research [*Catal Sci Technol*, 2012, 2,1977-1984; *Surf Interface Anal.* 2019,51,763–777.] do some semi-quantitative element content analysis based on XPS after parameter calibration and element sensitivity factor method. Here, we also note that there is not only a difference between Sn²⁺ and Cd²⁺, but also in terms of the intensity of Cl⁻, which should ideally be three times higher than that of Cs⁺ and Cd²⁺. However, if we compare with three times the intensity of Cs⁺ and Cd²⁺, the observed intensity of Cl⁻ is also significantly lower than anticipated. The similarity between Cd²⁺ and Cs⁺ also exhibits big differences. Regarding the different in intensity, it remains uncertain whether the XPS instrument has been appropriately calibrated for the relevant parameters of element content analysis (including element sensitivity factor, the ratio of signal to noise (S/N), function modification, etc.). However, it is worth emphasized that the instrument demonstrates feasibility for valence-state analysis and yields reliable results as the determination of element valence-state is unaffected by elemental content.

b) In the context of ICP-MS sample preparation, authors can dissolve the CsCdCl₃:10Sn²⁺ powder (ground the crystals to powder) in aqua regia and dilute

it in water.

Response: We thank the reviewer for the constructive comments. We benefit a lot from your suggestions and get relevant results as follow. The CsCdCl₃:10%Sn powder (ground the crystals to powder) can be dissolved in aqua regia and further be diluted in water. We can see that the not particularly large difference between actual Cd-to-Sn molar ratio value and corrected Vegard's law (XRD) or EDS values.

The relevant expression has been adjusted as below,

Page 6, line 156, one sentence has been added: “Furthermore, the ICP-OES value of the doping Sn²⁺ content in CsCdCl₃:10%Sn is 7.01% (Supplementary Table 4), which is similarly to the corrected Vegard's law (XRD) and EDS value.”

Supplementary Table 4. The result of ICP-OES measurements for CsCdCl₃:10%Sn.

Cd/Sn molar feed ratio	Actual Cd concentration (µg/mL)	Actual Sn concentration (µg/mL)	Actual Cd-to-Sn molar ratio
90.0% :10.0%	161.10	13.15	92.99% : 7.01%

c) It is suggested that the authors include images depicting the as-synthesized single crystals rather than SEM images. Notably, the SEM image in Fig. S6 (X=10) appears similar to polycrystals.

Response: We thank the reviewer for the effort on improving our work. According to the suggestion, we have added the Supplementary Information with additional images of the as-synthesized single crystals. Our respected reviewer, polycrystals can consist of both homogeneous or heterogeneous polycrystalline structures, we are very confused to use this term of polycrystals. First of all, we would like to understand the concept of polycrystals, which is composed of many single crystals (or small crystals), and they are essentially the same, only differing in arrangement orientation, with the same single-phase purity. If so, we can agree with the statement about polycrystals. If polycrystals refer to the composition of single crystals (or small crystals) that are completely different in nature, and are not single-phase purity, we will not use this polycrystals statement. As shown below (Supplementary Fig. 6-7), it is important to

clarify that not all crystals exhibit regular spindle shape structures, after all, there are instances of fractured crystals and polycrystals (single-phase purity). However, their luminous color remains remarkably consistent and pure, thereby demonstrating the uniformity of doping, corroborating the single-phase purity of the synthesized Sn^{2+} -doped CsCdCl_3 . We also covered this issue in the previous version that the powdered X-ray diffraction (PXRD) patterns of $\text{CsCdCl}_3:10\%\text{Sn}$ closely agree with the pattern in the PDF#18-0337 (Fig. 1. c), confirming the single-phase purity of the synthesized Sn^{2+} -doped CsCdCl_3 . In order to further solve the confusion, we also show the test sample cell below (Figure R3), in such a large sample analysis, if $\text{CsCdCl}_3:10\%\text{Sn}$ are impurities or crystals with different structures and different lattice parameter, it is impossible to get such a pure phase XRD peak. Therefore, we would like to employ the term "single-phase purity" to describe, but currently have not used the term "polycrystals", thereby avoiding this potential confusion. We hope the reviewer could understand this point.

The relevant expression has been adjusted as below,

Page 6, line 145, one sentence has been added: "From these images of as-synthesized crystals, all $\text{CsCdCl}_3:x\%\text{Br}$ and $\text{CsCdCl}_3:x\%\text{Sn}$ exhibit remarkably consistent and pure fluorescence colors (Supplementary Fig. 6-7), also demonstrating the uniformity and single-phase purity in this doping engineering."

Figure R3. XRD test sample cell

Fig. 1. c Rietveld refinements of the typical XRD patterns of $\text{CsCdCl}_3:10\%\text{Sn}$.

Supplementary Fig. 6. The fluorescence (under UV 254 nm light) and bright field images of $\text{CsCdCl}_3:x\%\text{Br}$, with **a** (0.2%Br), **b** (0.5%Br), **c** (0.8%Br), **d** (1%Br), **e** (3%Br), **f** (5%Br), **g** (10%Br) and **h** (15%Br).

Supplementary Fig. 7. The fluorescence (under UV 254 nm light) and bright field images of $\text{CsCdCl}_3:x\%\text{Sn}$, **a** (0%Sn), **b** (1%Sn), **c** (3%Sn), **d** (5%Sn), **e** (10%Sn) and **f** (15%Sn). Note that blue color is the reflection of the UV lamp.

d) The calculation of the Huang-Rhys factor indicates an increase in electron-phonon coupling with heightened Sn^{2+} dopant concentration. However, it remains ambiguous whether increased defects give rise to more STEs, thereby leading to persistent luminescence, or the charge carriers from the $5s^2$ orbital of Sn^{2+} contribute to the trap states to generate persistent luminescence.

Response: We thank the foresight and sagacity of reviewer for the suggestion to improve our manuscript. We apologize for the confusion caused by the unclear description to the reviewer. First, it is reasonable that 595 nm belongs to Cd, so we will not discuss it again. Next, we would like to further elaborate on the peak at 565 nm. We fully agree with your viewpoint that the charge carriers from the $5s^2$ orbital of Sn^{2+} contribute to the trap states to generate persistent luminescence. Specifically, the charge

carriers from the $5s^2$ orbital of Sn^{2+} , can be stored at the traps, when they are released and would migrate again to the luminescence center, which further coupled with the phonon to form the LPL of 565 nm. Lattice vibrations are present all the time and everywhere, so phonons also come from this. Under this electron-phonon coupling, the emission of 565 nm also has the properties of STE. The increase in electron-phonon coupling is attributed to the heightened concentration of Sn^{2+} dopants, which can be understood as the more amounts of the Sn center and the more opportunities for electron-phonon coupling, thereby increasing the Huang-Rhys factor.

The relevant expression has been adjusted as below,

Page 14, line 354, several sentences has been adjusted: “a) The lower excitation energy of 282 nm can effectively populate the charge carriers from the $5s^2$ orbital of Sn^{2+} into the lower-energy 3P_1 excited state, and further be saved by traps. These charge carriers also can escape from shallow traps or deep traps by thermodynamically breaking the energy barrier to forming the anti-thermal quenching ability.^{59-60,62} b) The charge carriers migrate back to the luminescent center and undergo electron-phonon coupling, resulting in the formation of LPL and STE of 565 nm. c) such high Huang-Rhys factor (S) (Supplementary Fig. 33a,b) and large Stokes shift crucially support STEs of 565 nm⁶³.”

At last, we wish to thank the Editor and the Reviewers again for the very constructive comments and suggestions to improve the quality of our manuscript. Thank you very much!

Reviewer #1 (Remarks to the Author):

The Author has revised accordingly and suggests acceptance of this manuscript.

Reviewer #2 (Remarks to the Author):

The authors addressed the raised points. From the theory point, the strengths and weaknesses of the project are clearly reported.

However, I am still skeptical on the added value of theory and how it complements the experimental results. Therefore, the main point of this paper is to establish if there is or not the experimental novelty.

Reviewer #3 (Remarks to the Author):

The authors have addressed comments and suggestions. The current version can be accepted as it is.

Reviewer #1 (Remarks to the Author):

The Author has revised accordingly and suggests acceptance of this manuscript.

Response: We sincerely appreciate the reviewer for acknowledgement of the acceptance of our work.

Reviewer #2 (Remarks to the Author):

The authors addressed the raised points. From the theory point, the strengths and weaknesses of the project are clearly reported.

However, I am still skeptical on the added value of theory and how it complements the experimental results. Therefore, the main point of this paper is to establish if there is or not the experimental novelty.

Response: We thank the reviewer for the valuable comment. We pay high attention to the reviewer's comment and will discuss the theoretical value and experimental novelty. With the development of science and technology, computational chemistry, as an important chemical research method, has been widely used in various fields [*Nature* 2024, 626, 517–522; *Science* 2024, 383, 86–93]. In the previous version, we have discussed in detail the crucial and necessary DFT calculation required to investigate the impact of Cl^- and Sn^{2+} on luminescence at different doping sites and symmetry centers. Experimental and theoretical calculations verify each other, showing that the relationship between fish and water cannot be separated, and we hope that our respected reviewer could understand this point.

In the mid-1990s, the initial report on $\text{SrAl}_2\text{O}_4:\text{Eu}^{2+}\text{-Dy}^{3+}$ gained significant attention due to its long-persistent luminescence (LPL) [*J. Electrochem. Soc.* 1996, 143, 2670]. Since then, purely inorganic systems with prolonged afterglow have continuously emerged across a wide range of optoelectronic applications, including optical displays, bioimaging, photonic technologies, and photocatalysis in modern society [*Nat. Mater.* 2023, 22, 289–304]. However, these purely inorganic materials have to undergo high-temperature synthesis (1000~1500 °C), which is not environmentally friendly and has safety risks. Faced with such a wide range of applications, and the vacancy of material upgrade iteration, Zhang *et al.* took the lead

in realizing the low temperature hydrothermal method for all-inorganic $\text{Cs}_2\text{Na}_x\text{Ag}_{1-x}\text{InCl}_6:y\%\text{Mn}$ perovskites with ultra-long afterglow in 2021 [*Angew. Chem. Int. Ed.* 2021, 60, 24450–24455]. In this emerging field, we firmly believe that there is a lot of challenging scientific work that is worth the effort and would arouse broader scholarly attention. This perspective is limited to purely inorganic materials, while the design and synthesis of long-afterglow materials have become a research hotspot for other materials, including both organic materials and inorganic-organic hybrid materials [*Chem. Soc. Rev.* 2021, 50, 5564; *Chem. Soc. Rev.* 2023, 52, 8005-8058; *Nature Reviews Materials* 2020, 5, 869–885; *Nature Reviews Chemistry* 2023, 7, 854–874]. Their afterglow lifetimes are mostly in the second level, and those exceeding a minute are still rarely reported. [*Nature* 2017, 550, 384–387; *Nature Photonics* 2024, 18, 350–356].

In this work, **a)** we get rid of high temperature dry synthesis, choose low temperature wet chemistry synthesis method to obtain all-inorganic halide perovskites with an afterglow duration time of over 2,000 s and a photoluminescence quantum yield of approximately 84.47%. **b)** Anti-thermal quenching properties, as the front research filed for preventing triplet exciton thermal quenching in room-temperature phosphorescence (RTP), LPL and optoelectronic materials [*Nat Commun* 2024, 15, 3598; *Nat Commun* 2020, 11, 4649. *J. Am. Chem. Soc.* 2022, 144, 2726–2734; *Angew. Chem.Int. Ed.* 2023,62, e20230917, etc.], where we have achieved the anti-thermal quenching temperature up to 377 K. **c)** Temperature-dependent phosphorescence, which is also a novel property for stimuli-responsive luminescent materials [*Nat Commun* 2024, 15, 2134; *Nat Commun*, 2024, 15,1269; *Nat Commun* 2023, 14, 627; *Nat Commun* 2021, 12, 1364; *Sci. Adv.* 2024, 10, eadm6928; *Adv. Mater.* 2023, 35, 2211; *Matter*, 2020, 3, 449; *Angew. Chem.Int. Ed.* 2022,61, e2022066; *Angew. Chem.Int. Ed.*, 2023,62, e2023027, etc.]. Here, $\text{CsCdCl}_3:0.8\%\text{Br}$ displays remarkable color variation with temperature, nearly full-color coverage, ranging from blue to cyan, then across yellow-green and finally to orange-red, observable with naked eyes. Such a wide range of full-color tunable luminescence and the anti-thermal quenching properties are still rare, particularly in state-of-the-art LPL materials. **d)** Given intelligent materials are capable of time-dependent phosphorescence, it will possess a bright future in high-class

security protection. This phenomenon is favored by researchers and has been reported as a significant performance for luminescent materials [*J. Am. Chem. Soc.* 2024, 146, 1294–1304; *Angew. Chem.Int. Ed.* 2023, 62, e20230306; *Angew. Chem.Int. Ed.* 2024, 63, e202316; *Adv. Mater.*2022, 34, 2108333. *Adv. Mater.*2020, 32, 2004768. *Adv. Funct. Mater.* 2023, 33, 2214962; *Adv. Funct. Mater.*2023, 33, 2208895, etc.]. Commonly, their color-varying afterglow is shifted from long to short wavelengths, while the occurrence of afterglow changing towards longer wavelengths is a rare and huge task. Importantly, to the best of our knowledge, the time-dependent afterglow reported so far is time-invariant and unregulated. At present, the time-valve controllable time-dependent afterglow is still a blank in the field. Frankly, we are the first to report a time-dependent color-changing afterglow with time-valve controllable character by Br⁻ ions doping system (Of course, due to journal requirements, we avoided using this term “first to report”). **e**) In 2019, excitation-wavelength-dependence (Ex-De) luminescence was described by Huang *et al.* as a valuable photophysical properties [*Nat. Photonics* 2019, 13, 406]. Since then, such materials have also been widely reported [*Nat Commun* 2023, 14, 8098; *Nat Commun* 2023, 14, 4163; *Nat Commun* 2022,13, 5712; *J. Am. Chem. Soc.* 2023, 145, 13392–13399; *J. Am. Chem. Soc.* 2022, 144, 12652–12660; *J. Am. Chem. Soc.* 2021, 143, 18317–18324; *Angew. Chem. Int. Ed.* 2023,62, e2023166; *Angew. Chem. Int. Ed.*2023,62, e2023166; *Angew. Chem. Int. Ed.*, 2020,59,6915 –6922; *Adv. Mater.* 2021, 33, 2007571; *Adv. Mater.* 2022, 34, 2204839; *Research.* 2021, 2021. etc.]. However, the vast majority of materials typically exhibit forward excitation-dependent luminescence, while the material with reverse excitation-dependent luminescence is intrinsically rare. In this work, CsCdCl₃:x%Sn as first one to exhibit forward and reverse excitation-dependent Janus-type luminescence, which is truly amazing for this property and emerges by lucky. **f**) these characteristics endow the LPL materials with dynamic tunability, offering new opportunities in high-security anti-counterfeiting and 5D information coding.

In a word, at present, looking at the entire searchable database from “Web of Science (Clarivate)” of state-of-the-art materials, it is difficult to find a class of materials similar to this work that can simultaneously achieve breakthroughs in

multiple properties, including anti-thermal quenching, temperature-, time-, and excitation-dependent luminescent characteristics. Therefore, we fully believe that this is an innovative and important work, which not only provides new concepts for the design and synthesis of such materials, but also will arouse extensive interest of readers. Finally, we also thank and hope that the respectful reviewer can see the time and efforts made by authors and all Editor and Reviewers to improve the quality of the manuscript.

Reviewer #3 (Remarks to the Author):

The authors have addressed comments and suggestions. The current version can be accepted as it is.

Response: We sincerely appreciate the reviewer for acknowledgement of the acceptance of our work.

At last, we wish to thank the Editor and the Reviewers again for the very constructive comments and suggestions to improve the quality of our manuscript. Thank you very much!